# Neurophysiological signatures of cortical micro-architecture

Golia Shafiei[1,2], Ben D. Fulcher [3], Bradley Voytek [4], Theodore D. Satterthwaite [2], Sylvain Baillet [1] & Bratislav Misic [1] ✉

Systematic spatial variation in micro-architecture is observed across the cortex. These micro-architectural gradients are reflected in neural activity, which can be captured by neurophysiological time-series. How spontaneous neurophysiological dynamics are organized across the cortex and how they arise from heterogeneous cortical micro-architecture remains unknown. Here we extensively profile regional neurophysiological dynamics across the human brain by estimating over 6800 time-series features from the resting state magnetoencephalography (MEG) signal. We then map regional time-series profiles to a comprehensive multi-modal, multi-scale atlas of cortical micro-architecture, including microstructure, metabolism, neurotransmitter receptors, cell types and laminar differentiation. We find that the dominant axis of neurophysiological dynamics reflects characteristics of power spectrum density and linear correlation structure of the signal, emphasizing the importance of conventional features of electromagnetic dynamics while identifying additional informative features that have traditionally received less attention. Moreover, spatial variation in neurophysiological dynamics is co-localized with multiple micro-architectural features, including gene expression gradients, intracortical myelin, neurotransmitter receptors and transporters, and oxygen and glucose metabolism. Collectively, this work opens new avenues for studying the anatomical basis of neural activity.

Signals, in the form of electrical impulses, are perpetually generated, propagated, and integrated via multiple types of neurons and neuronal populations[1,2]. The wiring of the brain guides the propagation of signals through networks of nested polyfunctional neural circuits[3,4]. The resulting fluctuations in membrane potentials and firing rates ultimately manifest as patterned neurophysiological activity[5–7].

A rich literature demonstrates links between cortical micro-architecture and dynamics. Numerous studies have investigated the cellular and laminar origins of cortical rhythms[8–13]. For instance, electro- and magneto-encephalography (EEG/MEG) signals appear to be more sensitive to dipoles originating from pyramidal cells of cortical layers II-III and V[14,15]. Moreover, specific time-series features of neuronal electrophysiology depend on neuron type, morphology and local gene transcription, particularly genes associated with ion channel regulation[16–18]. However, previous studies have mostly focused on single or small sets of features-of-interest, often mapping single micro-architectural features to single dynamical features. Starting with the discovery of 8–12 Hz alpha rhythm in the electroencephalogram[19], conventional time-series analysis in neuroscience has typically focused on canonical electrophysiological rhythms[20–24]. More recently, there has also been a growing interest in studying the intrinsic timescales that display a hierarchy of temporal processing from fast fluctuating

[1]McConnell Brain Imaging Centre, Montréal Neurological Institute, McGill University, Montréal, Canada. [2]Department of Psychiatry, Perelman School of Medicine, University of Pennsylvania, Philadelphia, PA 19104, USA. [3]School of Physics, The University of Sydney, Camperdown, NSW 2006, Australia. [4]Department of Cognitive Science, Halıcıoğlu Data Science Institute, University of California, San Diego, La Jolla, CA, USA. ✉e-mail: bratislav.misic@mcgill.ca

activity in unimodal cortex to slower encoding of contextual information in transmodal cortex[25–34]. How ongoing neurophysiological dynamics arise from specific features of neural circuit microarchitecture remains a key question in neuroscience[1,2,12].

Recent analytic advances have opened new opportunities to perform neurophysiological time-series phenotyping by computing comprehensive feature sets that go beyond power spectral measures, including measures of signal amplitude distribution, entropy, fractal scaling and autocorrelation[35–40]. Concomitant advances in imaging technologies and data sharing offer new ways to measure brain structure with unprecedented detail and depth[41–43], including gene expression[44], myelination[45,46], neurotransmitter receptors[47–54], cytoarchitecture[55–57], laminar differentiation[56,58], cell type composition[44,59,60], metabolism[61,62] and evolutionary expansion[63,64].

Here we comprehensively characterize the dynamical signature of neurophysiological activity and relate it to the underlying microarchitecture by integrating multiple, multimodal maps of human cortex. Instead of manually selecting a small number of features-of-interest, we use extensive sets of dynamical and micro-architectural features using data-driven approaches. Specifically, we first derive cortical spontaneous neurophysiological activity using source-resolved magnetoencephalography (MEG) from the Human Connectome Project (HCP; see ref. 65). We then apply highly comparative time-series analysis (hctsa; see refs. 35,36) to estimate a comprehensive set of time-series features for each brain region (Fig. 1). At the same time, we construct a micro-architectural atlas of the cortex that includes maps of microstructure, metabolism, neurotransmitter receptors and transporters, laminar differentiation and cell types (Fig. 2). Finally, we map these extensive micro-architectural and dynamical atlases to one another using multivariate statistical analysis.

## Results

Regional neurophysiological time-series were estimated by applying linearly constrained minimum variance (LCMV) beamforming to resting state MEG data from the Human Connectome Project (HCP; see ref. 65) using Brainstorm software[66] (see *Methods* for details). Highly comparative time-series analysis (hctsa; see refs. 35,36) was then used to perform massive time-series feature extraction from regional MEG recordings. This procedure provides a feature-based representation of time-series, where given time-series are represented by time-series feature vectors[36,37]. This time-series phenotyping analysis is a data-driven method that quantifies dynamic repertoire of neural activity using interdisciplinary metrics of

temporal structure of the signal and yields a comprehensive 'fingerprint' of dynamical properties of each brain region. Applying time-series phenotyping to regional MEG time-series, we estimated 6880 time-series features for 100 cortical regions from the Schaefer-100 atlas[67]. The hctsa library contains a vast and interdisciplinary set of features with potentially correlated values that span various conceptual time-series characteristics. The list of time-series features includes, but is not limited to, statistics derived from the autocorrelation function, power spectrum, amplitude distribution, and entropy estimates (Fig. 1).

To estimate a comprehensive set of multimodal microarchitectural features, we used the recently-developed neuromaps toolbox[43] as well as the BigBrainWarp toolbox[57], the Allen Human Brain Atlas (AHBA[44]) and the abagen toolbox[68] to transform and compile a set of 45 features, including measures of microstructure, metabolism, cortical expansion, receptors and transporters, layer thickness and cell type-specific gene expression (Fig. 2). Note that the microstructure maps include principal gradients of gene expression and neurotransmitter profiles as they each represent proxy measures of certain molecular properties. Specifically, the principal component of gene expression (gene expression PC1) provides a potential proxy for cell type distribution across the cortex[44,69,70] and the principal component of neurotransmitter receptors and transporters (neurotransmitter PC1) provides a summary measure of protein densities of 18 neurotransmitter receptors and transporters[47,71]. We also included individual neurotransmitter receptor and transporter maps as well as cell type-specific gene expression maps to assess their effects separately.

In subsequent analyses, we first assess the topographic organization of neurophysiological dynamics by quantifying the dominant patterns of variations in resting-state MEG time-series properties. We then characterize the signature of neurophysiological dynamics with respect to micro-architectural attributes across the cortex. Finally, we perform sensitivity analyses to investigate potential effects of confounding factors on the findings, such as signal-to-noise ratio and parcellation resolution (see *Sensitivity analysis* for details).

### Topographic distribution of neurophysiological dynamics

The hctsa time-series phenotyping procedure generated 6880 time-series features per brain region. Since hctsa contains multiple algorithmic variants for quantifying any given time-series property, the identified time-series features potentially capture related dynamical behaviour and include groups of correlated properties. Hence, we first

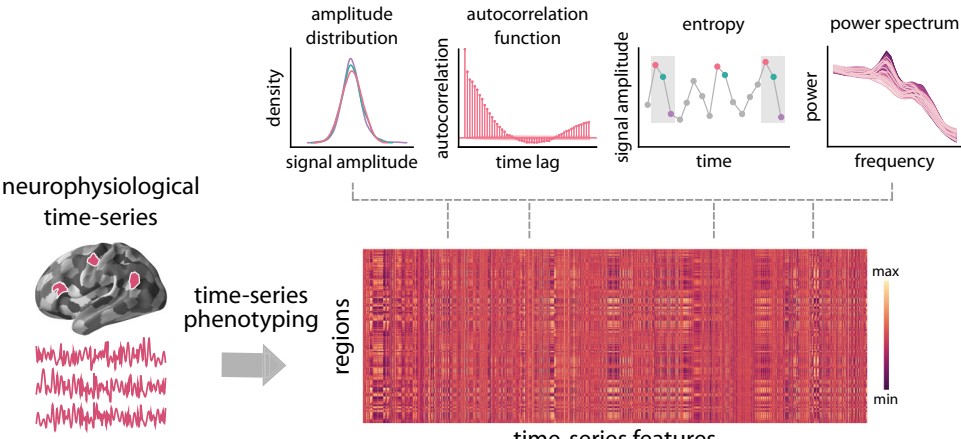

**Fig. 1 | Feature-based representation of neurophysiological time-series.** Highly comparative time-series analysis (hctsa; see ref. 35) toolbox was used to perform time-series feature extraction on regional MEG time-series. This time-series phenotyping procedure generated 6 880 time-series features for each region, including measures of autocorrelation, entropy, power spectrum and amplitude distribution. Source data are provided as a Source Data file.

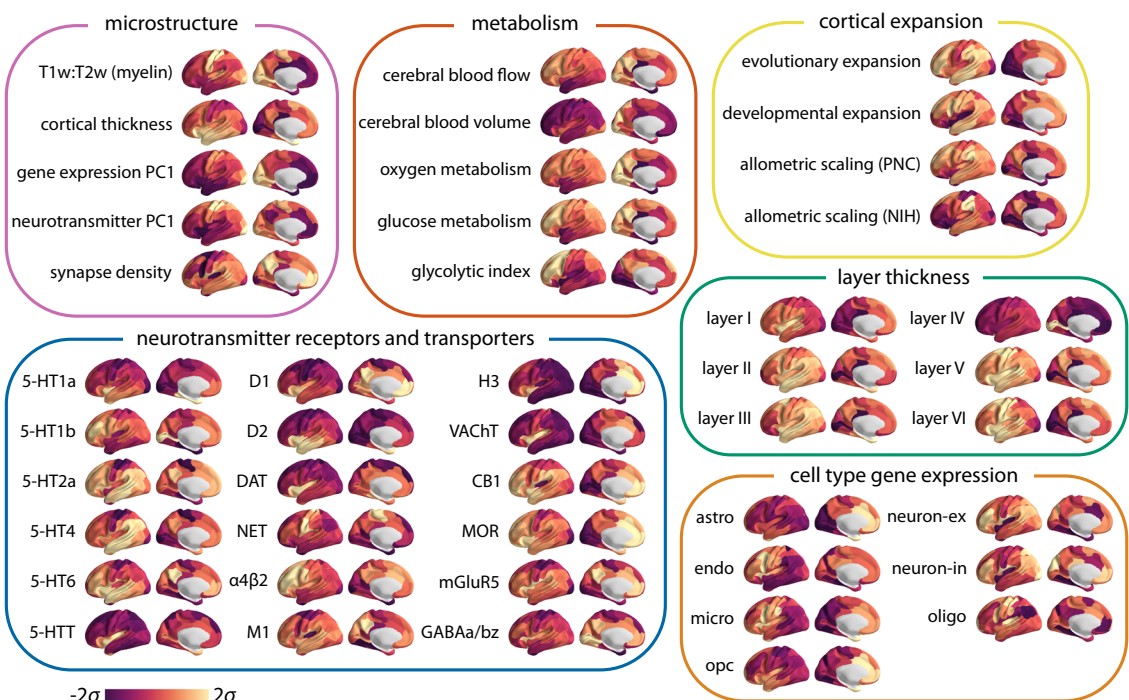

**Fig. 2 | Multimodal brain maps.** `neuromaps` toolbox[43], `BigBrainWarp` toolbox[57], the Allen Human Brain Atlas (AHBA[44]) and the `abagen` toolbox[68] were used to compile a set of 45 micro-architectural brain maps, including measures of microstructure, metabolism, cortical expansion, receptors and transporters, layer thickness and cell type-specific gene expression (astro = astrocytes; endo = endothelial cells; micro = microglia; neuron-ex = excitatory neurons; neuron-in = inhibitory neurons; oligo = oligodendrocytes; and opc = oligodendrocyte precursors)

(see *Methods* for more details). Note that the microstructure maps include principal gradients of gene expression and neurotransmitter profiles, for which we have also separately included feature sub-sets (specific receptor maps and cell type-specific gene expression). All obtained brain maps are depicted across the cortex at 95% confidence interval (Schaefer-100 atlas[67]). Source data are provided as a Source Data file.

sought to identify dominant macroscopic patterns or gradients of neurophysiological dynamics using principal component analysis (PCA)[38]. Applying PCA to the group-average region × feature matrix, we find evidence of a single dominant component that captures 48.7% of the variance in regional time-series features (Fig. 3a). The dominant component or "gradient" of neurophysiological dynamics (PC1) mainly spans the posterior parietal cortex and sensory-motor cortices on one end and the anterior temporal, orbitofrontal and ventromedial cortices on the other end (Fig. 3a). Focusing on intrinsic functional networks, we find that the topographic organization of the dominant neurophysiological dynamics varies along a sensory–fugal axis from dorsal attention, somatomotor and visual networks to limbic and default mode networks[72] (Fig. 3a).

We next investigated the top-loading time-series features on the first component, using the univariate correlations between each of the original feature maps and the PC1 map (i.e., PCA loadings). All correlations were statistically assessed using spatial autocorrelation-preserving null models ("spin tests"[73,74]; see *Methods* for details). Figure 3b shows that numerous features are positively and negatively correlated with PC1; the full list of features, their correlation coefficients and p-values are available in the online Supplementary Dataset S1. Inspection of the top loading features reveals that the majority are statistics derived from the structure of the power spectrum or closely related measures. Examples include power in different frequency bands, parameters of various model fits to the power spectrum, and related measures, such as the shape of the autocorrelation function and measures of fluctuation analysis. Figure 3b shows how the power spectrum varies across the cortex, with each line representing a brain region. Regions are coloured by their position in the putative unimodal–transmodal hierarchy[75]; the variation visually suggests that

unimodal regions display more prominent alpha (8–12 Hz) and beta (15–29 Hz) power peaks. Collectively, these results demonstrate that the traditional focus of electrophysiological time-series analysis on statistics of the power spectrum is consistent with the dominant variations in MEG dynamics captured by the diverse library of `hctsa` time-series features.

Given that the topographic organization of PC1 was closely related to power spectral features, we directly tested the link between PC1 and conventional band-limited power spectral measures[21–23], as well as intrinsic timescale[30] (Supplementary Figure 1). Figure 3c shows the correlations between PC1 and delta (2–4 Hz), theta (5–7 Hz), alpha (8–12 Hz), beta (15–29 Hz), lo-gamma (30–59 Hz) and hi-gamma (60–90 Hz) power maps, and intrinsic neural timescale[28,30–34,76]. We find that PC1 is significantly correlated with intrinsic timescale ($r_s = 0.84$, $p_{spin} = 0.038$; FDR-corrected) and hi-gamma ($r_s = 0.87$, $p_{spin} = 0.006$; FDR-corrected). The results were consistent when we used band-limited power maps that were adjusted for the aperiodic component of the power spectrum as opposed to the total power[21] (Supplementary Fig. 2). The fact that PC1 correlates with intrinsic timescale is consistent with the notion that both capture broad variations in the power spectrum. Given that the intrinsic timescale reflects characteristics of the aperiodic component of the power spectrum (these measures are mathematically related; see *Methods* for details), we also directly assessed the association between PC1 and the exponent and offset of the aperiodic component. The exponent describes the "curve" or the overall "line" or the slope of the aperiodic component and the offset describes the overall vertical shift (up and down translation) of the whole power spectrum[21]. PC1 was significantly correlated with both measures, suggesting that time-series features captured by PC1 also reflect properties of the aperiodic component of

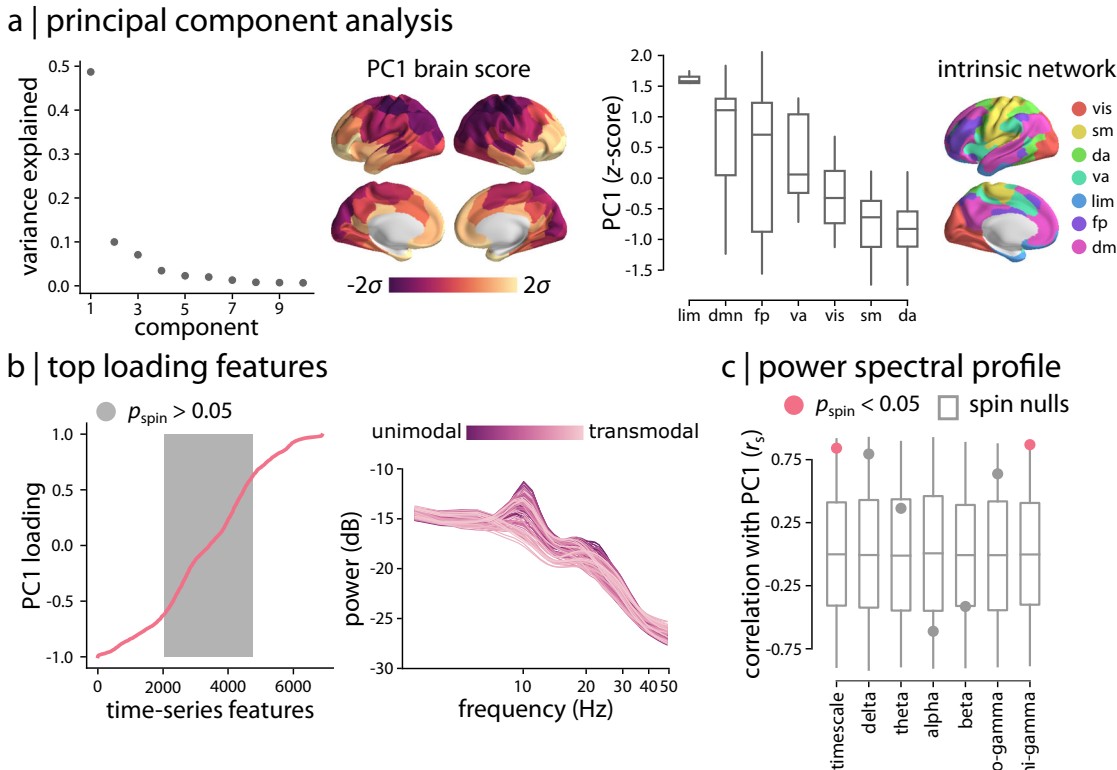

**Fig. 3 | Topographic distribution of neurophysiological dynamics. a** Principal component analysis (PCA) was used to identify linear combinations of MEG time-series features with maximum variance across the cortex. The first principal component (PC1) accounts for 48.7% of the total variance in neurophysiological time-series features. The spatial organization of the dominant time-series features captured by PC1 is depicted across the cortex at 95% confidence interval. The distribution of PC1 brain score is also depicted for intrinsic functional networks[72]. Centre line of the box plots represents the median, whiskers represent the minima and maxima, and bounds represent the 1st (25%) and 3rd (75%) quartiles of the distribution. Number of brain regions in each intrinsic network is: $N_{lim} = 5$, $N_{dmn} = 24$, $N_{fp} = 13$, $N_{va} = 12$, $N_{vis} = 17$, $N_{sm} = 14$, $N_{da} = 15$. **b** To examine the feature composition of the time-series features captured by PC1, feature loadings were estimated as the correlation coefficients between each `hctsa` time-series feature and PC1 brain score. PC1 loadings are depicted for the time-series features (ordered by their individual loadings). Grey background indicates non-significant features based on 10,000 spatial autocorrelation-preserving permutation tests (i.e., "spin tests"[73,74]; two-tailed; corrected for multiple comparisons using FDR correction).

The top loading features were mainly related to the power spectrum of regional time-series and its structure. Regional power spectral densities are depicted, with each line representing a brain region. Regions are coloured by their position in the putative unimodal–transmodal hierarchy[75]. **c** To contextualize the principal component of variation in MEG time-series features, PC1 brain score was correlated with MEG power maps at 6 canonical frequency bands and intrinsic timescale. The observed correlations are shown by filled circles and are compared to their corresponding null distributions of correlations obtained from 10,000 spatial autocorrelation-preserving permutation tests ("spin nulls" depicted as grey box plots). PC1 score is significantly correlated with intrinsic timescale ($r_s = 0.84$, $p_{spin} = 0.038$) and hi-gamma power ($r_s = 0.87$, $p_{spin} = 0.006$) (two-tailed; FDR-corrected). $r_s$ denotes the Spearman's rank correlation coefficient. Centre line of the box plots represents the median, whiskers represent the minima and maxima, and bounds represent the 1st (25%) and 3rd (75%) quartiles of the distribution. $N_{regions} = 100$ for each box plot. Intrinsic networks: vis visual, sm somatomotor, da dorsal attention, va ventral attention, lim limbic, fp frontoparietal, dmn default mode. Source data are provided as a Source Data file.

the power spectrum (Supplementary Fig. 3). In addition, previous reports suggest that broadband gamma activity also partly reflects the aperiodic neurophysiological activity and broadband shifts in the power spectrum[77–79]. This is consistent with our findings that PC1 is associated with gamma power and intrinsic timescale, mainly capturing broad variations in power spectrum and characteristics of the aperiodic activity.

Note that we focused on PC1 because the other components (PC2 and above) accounted for 10% or less of the variance in time-series features and were not significantly associated with `hctsa` time-series features. Moreover, to verify that the apparent low-dimensionality of the data and the identified PC patterns were not driven by the smaller number of samples (i.e., brain regions) than features (i.e., time-series features), we performed a sensitivity analysis where we randomly selected 100 time-series features (from the original list of 6 880 features) and re-ran PCA (1 000 repetitions). The identified PC patterns and their corresponding amount of variance explained were consistent with the original analysis using the full set of time-series features (Supplementary Fig. 4).

## Neurophysiological signatures of micro-architecture

How do the regional neurophysiological time-series features map onto multimodal micro-architectural features? To address this question, we implemented a multivariate partial least squares analysis (PLS; see refs. [80,81]) that integrates multiple multimodal brain maps into the analysis and seeks to identify linear combinations of time-series features and linear combinations of micro-architectural features that optimally covary with one another. Figure 4a shows that the analysis identifies multiple such combinations, termed latent variables (similar results were obtained using sparse canonical correlation analysis (sCCA); Supplementary Fig. 5). Statistical significance of each latent variable was assessed using spatial autocorrelation-preserving permutation tests[70,74]. The first latent variable was statistically significant, capturing the greatest covariance between time-series and micro-architectural features (covariance explained = 75.4%, $p_{spin} = 0.011$).

Figure 4b shows the spatial topography of time-series features and micro-architectural scores for the first latent variable. These are the weighted sums of the original input features according to the weighting identified by the latent variable. The correlation between the

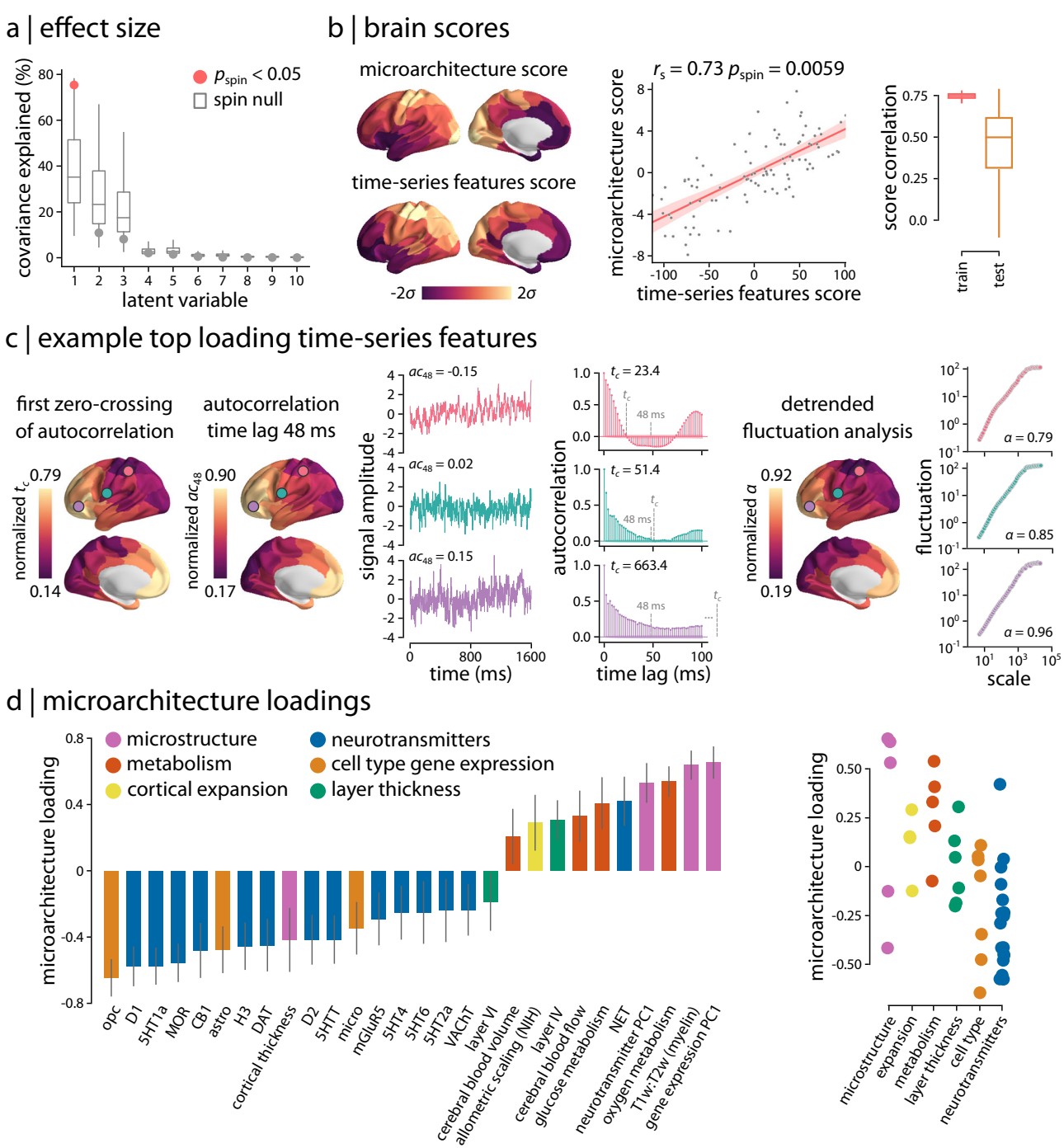

**a | effect size**

**b | brain scores**

microarchitecture score

time-series features score

$r_s = 0.73$ $p_{spin} = 0.0059$

**c | example top loading time-series features**

first zero-crossing of autocorrelation

autocorrelation time lag 48 ms

$ac_{48} = -0.15$

$ac_{48} = 0.02$

$ac_{48} = 0.15$

$t_c = 23.4$

$t_c = 51.4$

$t_c = 663.4$

detrended fluctuation analysis

$\alpha = 0.79$

$\alpha = 0.85$

$\alpha = 0.96$

**d | microarchitecture loadings**

microstructure · neurotransmitters · metabolism · cell type gene expression · cortical expansion · layer thickness

score maps is maximized by the analysis ($r_s = 0.73$, $p_{spin} = 0.0059$). We therefore sought to estimate whether the same mapping between time-series and micro-architectural features can be observed out-of-sample. We adopted a distance-dependent cross-validation procedure where "seed" regions were randomly chosen and the 75% most physically proximal regions were selected as the training set, while the remaining 25% most physically distal regions were selected as the test set[70] (see *Methods* for more details). For each train-test split, we fit a PLS model to the train set and project the test set onto the weights (i.e., singular vectors) derived from the train set. The resulting test set scores are then correlated to estimate an out-of-sample correlation coefficient. Figure 4b shows that micro-architecture and time-series feature scores are correlated in training set (mean $r_s = 0.75$) and test set (mean $r_s = 0.5$), demonstrating consistent findings in out-of-sample analysis.

We next examined the corresponding time-series and micro-architecture feature loadings and identified the most contributing features to the spatial patterns captured by the first latent variable (Fig. 4c, d). The top loading time-series features were mainly related to measures of self-correlation or predictability of the MEG signal. The self-correlation measures mostly reflect the linear correlation structure of neurophysiological time-series, particularly long-lag autocorrelations (at lags > 15 time steps, or > 30 ms). A wide range of other highly weighted time-series features captured other aspects of signal predictability, including measures of the shape of the autocorrelation function (e.g., the time lag at which the autocorrelation function crosses zero), how the autocorrelation structure changes after removing low-order local trends (e.g., residuals from fitting linear models to rolling 5-time-step, or 10 ms, windows), scaling properties assessed using fluctuation analysis (e.g., scaling of signal variance

**Fig. 4 | Neurophysiological signature of micro-architecture. a** Partial least square (PLS) analysis was used to assess the multivariate relationship between micro-architectural and time-series features. PLS identified a single significant latent variable (covariance explained = 75.4%, $p_{spin}$ = 0.011, two-tailed). Centre line of the box plots represents the median, whiskers represent the minima and maxima, and bounds represent the 1st (25%) and 3rd (75%) quartiles of the distribution. Number of observations (i.e., null covariance explained) for each box plot is $N$ = 10000 (number of applied spin tests). **b** Spatial patterns of micro-architecture and time-series features scores are depicted for the first latent variable. The two brain score maps are significantly correlated ($r_s$ = 0.73, $p_{spin}$ = 0.0059, two-tailed). $r_s$ denotes the Spearman's rank correlation coefficient; linear regression line is added to the scatter plot for visualization purposes only (shaded area denotes 95% confidence interval for the regression). To assess the out-of-sample correlation of brain scores, a distance-dependent cross-validation analysis was used (see *Methods*). Micro-architecture and time-series feature scores are consistently correlated in both training set (75% of regions; mean $r_s$ = 0.75) and test set (25% of regions; mean $r_s$ = 0.5). Centre line of the box plots represents the median, whiskers represent the minima and maxima, and bounds represent the 1st (25%) and 3rd (75%) quartiles of the distribution. Number of observations (i.e., score correlation) for each box plot is $N$ = 99 (number of applied train-test splits). **c** Top loading time-series features were mainly related to measures of self-correlation or predictability of the signal. Three examples of top loading features are depicted across the cortex. Left: first zero-crossing time point of the autocorrelation function, $t_c$, and linear auto-correlation at a lag of 48 ms, $ac_{48}$; right: the scaling exponent of detrended fluc-tuation analysis, $\alpha$. Short segments of raw time-series, autocorrelation functions, and fluctuation analysis plots (log-log plot of detrended fluctuations at multiple timescales) are also shown for a randomly selected participant at three cortical regions (circles on the brain surface: pink ≈ 5th percentile, green ≈ 50th percentile, purple ≈ 95th percentile). Time points corresponding to zero-crossing point ($t_c$) and 48 ms are indicated with grey dashed lines on the autocorrelation function plots. **d** PLS loadings for micro-architectural features are shown for each set of brain maps as bar charts (left; only reliable loadings are shown) and scatter plots (right). Bar charts depict PLS loadings (a single PLS loading per micro-architectural map) with their corresponding 95% confidence intervals from 10,000 bootstrap resamplings. Cell types: astro astrocytes, micro microglia, opc oligodendrocyte precursors. Source data are provided as a Source Data file.

across timescales), and measures derived from a wavelet decomposi-tion (e.g., wavelet coefficients at different timescales). The full list of time-series feature loadings for the first latent variable is available in the online Supplementary Dataset S2.

To illustrate the spatial distribution of highly contributing time-series features, Fig. 4c shows three top-loading features that mirror the spatial variation of the first PLS latent variable. For example, Figure 4c, *left* depicts the distribution of the group-average first zero-crossing point of the autocorrelation function. The autocorrelation function of the unimodal cortex (marked with a pink circle) crosses zero auto-correlation at a lower lag than the transmodal cortex (marked with a purple circle), suggesting faster autocorrelation decay and longer correlation length in transmodal cortex than in unimodal cortex. Another example is the linear autocorrelation of the MEG signal at longer time lags. Figure 4c, *left* shows autocorrelation at a lag of 48 ms (24 time steps), demonstrating lower autocorrelation in unimodal cortex and higher autocorrelation in transmodal cortex. Note that the list of top-loading features includes linear autocorrelation at other time lags and autocorrelation at a lag of 48 ms was only selected as an illustrative example (Supplementary Fig. 6 depicts the full range of loadings for linear autocorrelation at all time lags included in `hctsa`). Finally, we examined the scaling exponent, $\alpha$, estimated using detrended fluctuation analysis as the slope of a linear fit to the log-log plot of the fluctuations of the detrended signal across timescales[82,83]. Figure 4c, *right* depicts this scaling exponent across the cortex, which exhibits a similar spatial pattern as the previous two examples, indi-cating lower self-correlation in unimodal cortex (pink circle) compared to transmodal cortex (purple circle). Other variations of fluctuation analysis also featured heavily in the list of top-loading features, including goodness of fit of the linear fit, fitting of multiple scaling regimes, and different types of detrending and mathematical for-mulation of fluctuation size.

Figure 4d shows the corresponding micro-architectural loadings. The most contributing micro-architectural features to the spatial pat-terns captured by the first latent variable are the principal component of gene expression (gene expression PC1; a potential proxy for cell type distribution[44,69,70]), T1w/T2w ratio (a proxy for intracortical myelin[46]), principal component of neurotransmitter receptors and transporters (neurotransmitter PC1), and oxygen and glucose meta-bolism (strong positive loadings). We also find high contributions (strong negative loadings) from specific neurotransmitter receptor and transporters, in particular metabotropic serotonergic and dopa-minergic receptors, as well as from cell type-specific gene expression of oligodendrocyte precursors (opc), which are involved in myelinogenesis[84–88]. Consistent findings were obtained when we used univariate analysis to relate regional time-series features and the top

loading micro-architectural maps, in particular principal component of gene expression and T1w/T2w ratio, which have previously been extensively studied as archetypical micro-architectural gradients[30,41,69,89,90] (Supplementary Fig. 7). Altogether, this analysis provides a comprehensive chart or 'lookup table' of how micro-architectural and time-series feature maps are associated with one another. These results demonstrate that cortical variation in multiple micro-architectural attributes manifests as a gradient of time-series properties of neurophysiological activity, particularly the properties that reflect the long-range self-correlation structure of the signal.

## Sensitivity analysis

To assess the extent to which the results are affected by potential confounding factors and methodological choices, we repeated the analyses using alternative approaches. First, to ensure that the findings are not influenced by MEG signal-to-noise ratio (SNR), we calculated SNR at each source location using a noise model that estimates how sensitive the source-level MEG signal is to source location and orientation[91,92]. We performed two follow-up analyses using the SNR map (Supplementary Fig. 8): (1) SNR was first compared with the full set of MEG time-series features using mass univariate Pearson corre-lations. Time-series features that were significantly correlated with SNR were removed from the feature set without correcting for multi-ple comparisons ($p_{spin}$ < 0.05; 10,000 spatial autocorrelation-preserving permutation tests[73,74]). Note that this is a more con-servative feature selection procedure compared to conventional multiple comparisons correction, because fewer features would be removed if correction for multiple comparisons was applied. PCA was applied to the remaining set of features (Supplementary Figure 8b). The principal component of the retained 3819 features (i.e., PC1 - feature subset) explained 31.6% of the variance and was significantly correlated with the original PC1 of the full set of features ($r_s$ = 0.93, $p_{spin}$ = 0.0001), reflecting similar spatial pattern as the original analy-sis. (2) SNR was regressed out from the full set of time-series features using linear regression analysis. PCA was then applied to the resulting feature residuals (Supplementary Fig. 8c). The principal component of SNR-regressed features (i.e., PC1 - SNR regressed) explained 41.4% of the variance and reflected the same spatial pattern as the original analysis ($r_s$ = 0.70, $p_{spin}$ = 0.0004). Moreover, we assessed the effects of environmental and instrumental noise on the findings, where we applied principal component analysis to the `hctsa` features obtained from pre-processed empty-room MEG recordings[23] (see *Methods* for more details). PCA weights of the time-series features of the empty-room MEG recordings were aligned with the PCA weights of the time-series features of the resting-state MEG recordings using the Pro-crustes method (see ref. 93; https://github.com/satra/mapalign). The

principal component of neurophysiological dynamics was then compared with the principal component of time-series features obtained from empty-room recordings, where no significant associations were identified (Supplementary Fig. 9; $r_s = -0.17$, $p_{spin} = 0.69$). These analyses demonstrate that the time-series features captured by the dominant axis of variation in neurophysiological dynamics are independent from measures of MEG signal-to-noise ratio.

Finally, to ensure that the findings are independent from the parcellation resolution, we repeated the analyses using a higher resolution parcellation (Schaefer-400 atlas with 400 cortical regions[67]). The results were consistent with the original analysis (Supplementary Figs. 10, 11). In particular, the first principal component (PC1) accounted for 48.6% of the variance and displayed a similar spatial organization as the one originally obtained for the Schaefer-100 atlas (Supplementary Fig. 10a). As before, the top loading time-series features were mainly related to the characteristics of the power spectral density (Supplementary Fig. 10b, c). The full list of features, their loadings and $p$-values are available in the online Supplementary Dataset S3. Moreover, PLS analysis identified a single significant latent variable ($p_{spin} = 0.0083$) that accounted for 75.7% of the covariance (Supplementary Fig. 11a). Micro-architecture and time-series feature scores displayed similar spatial patterns to the ones obtained for the Schaefer-100 atlas (Supplementary Fig. 11b). The corresponding feature loadings were also consistent with the original findings (micro-architectural loadings in Supplementary Fig. 11c and time-series feature loadings in the online Supplementary Dataset S4.)

## Discussion

In the present study, we use time-series phenotyping analysis to comprehensively chart the dynamic fingerprint of neurophysiological activity from the resting-state MEG signal. We then map the resulting dynamical atlas to a multimodal micro-architectural atlas to identify the neurophysiological signatures of cortical micro-architecture. We demonstrate that cortical variation in neurophysiological time-series properties mainly reflects power spectral density and is closely associated with intrinsic timescale and self-correlation structure of the signal. Moreover, the spatial organization of neurophysiological dynamics follows gradients of micro-architecture, such as neurotransmitter receptor and transporters, gene expression and T1w/T2w ratios, and reflects cortical metabolic demands.

Numerous studies have previously investigated neural oscillations and their relationship with neural communication patterns in the brain[8,10,11,94]. Previous reports also suggest that neural oscillations influence behaviour and cognition[94-98] and are involved in multiple neurological diseases and disorders[97,99]. Neural oscillations manifest as the variations of power amplitude of neurophysiological signal in the frequency domain[10,21,100,101]. Power spectral characteristics of the neurophysiological signal, such as mean power amplitude in canonical frequency bands, have previously been used to investigate the underlying mechanisms of large-scale brain activity and to better understand the individual differences in brain function[22,23,31,98,102,103]. Other time-series properties that are related to the power spectral density have also been used to study neural dynamics, including measures of intrinsic timescale and self-affinity or self-similarity of the signal (e.g., autocorrelation and fluctuation analysis)[25,30,82,83,104-106].

Applying a data-driven time-series feature extraction analysis, we find that the topographic organization of neurophysiological time-series signature follows a sensory–fugal axis, separating somatomotor, occipital and parietal cortices from anterior temporal, orbitofrontal and ventromedial cortices. This dynamic fingerprint of neurophysiological activity is mainly characterized by linear correlation structure of MEG signal captured by `hctsa` time-series features. The linear correlation structure manifests in both power spectral properties and the autocorrelation function. This dominant spatial variation of time-series features also resembles the spatial distribution of intrinsic

timescale, another measure related to the characteristics of power spectral density[28,30,33]. Altogether, while the findings highlight underrepresented time-series features, they emphasize the importance of conventional methods in characterizing neurophysiological activity and the key role of linear correlation structure in MEG dynamics.

Earlier reports found that regional neural dynamics, including measures of power spectrum and intrinsic timescale, reflect the underlying circuit properties and cortical micro-architecture[25,28,30]. The relationship between neural dynamics and cortical micro-architecture is often examined using a single, or a few microstructural features. Recent advances in data collection and integration and the increasing number of data sharing initiatives have provided a unique opportunity to comprehensively study cortical circuit properties and micro-architecture using a wide range of multimodal datasets[43,44,47,56,57,65,107]. Here we use such datasets and compile multiple micro-architectural maps, including measures of microstructure, metabolism, cortical expansion, receptors and transporters, layer thickness and cell type-specific gene expressions, to chart the multivariate associations between neurophysiological dynamics and cortical micro-architecture.

Our findings build on previous reports by showing that neurophysiological dynamics follow the underlying cytoarchitectonic and microstructural gradients. In particular, our findings confirm that MEG intrinsic dynamics are associated with the heterogeneous distribution of gene expression and intracortical myelin[30,89,108,109] and neurotransmitter receptors and transporters[47]. In addition, we link the dynamic signature of ongoing neurophysiological activity with multiple metabolic attributes[62,110]; for instance, we find that regions with greater oxygen and glucose metabolism tend to display lower temporal autocorrelation and therefore more variable moment-to-moment intrinsic activity. This is consistent with previously reported high metabolic rates of oxygen and glucose consumption in the sensory cortex[61]. We also find a prominent association with cell type-specific gene expression of oligodendrocyte precursors (opc), potentially reflecting the contribution of these cells to myelin generation by giving rise to myelinating oligodendrocytes during development[84-88] and to myelin regulation and metabolic support of myelinated axons in the adult neural circuits[87,88,111]. Finally, we find that the dominant dynamic signature of neural activity covaries with the granular cortical layer IV, consistent with the idea that layer IV receives prominent subcortical (including thalamic) feedforward projections[112,113]. Collectively, our findings build on the emerging literature on how heterogeneous micro-architectural properties along with macroscale network embedding (e.g., cortico-cortical connectivity and subcortical projections) jointly shape regional neural dynamics[38-40,114-117].

The present findings must be interpreted with respect to several methodological considerations. First, we used MEG data from a subset of individuals with no familial relationships from the HCP dataset. Although all the presented analyses are performed using the group-level data, future work with larger sample sizes can provide more generalizable outcomes[118,119]. Larger sample sizes will also help go beyond associative analysis and allow for predictive analysis of neural dynamics and micro-architecture in unseen datasets. Second, MEG is susceptible to low SNR and has variable sensitivity to neural activity from different regions (i.e., sources). Thus, electrophysiological recordings with higher spatial resolution, such as intracranial electroencephalography (iEEG and ECoG), may provide more precise measures of neural dynamics that can be examined with respect to cortical micro-architecture. However, a major caveat with iEEG and ECoG is that they lack whole brain coverage, limiting their practical usage in such analysis. An alternative non-invasive modality is on-scalp MEG, which offers both high SNR and spatial resolution[120-123]. Third, we note that the included micro-architectural maps are by no means direct measurements of the underlying neurobiological features. For example, the "myelin" map is estimated based on the ratio of T1-weighted to

T2-weighted MRI scans, which is only sensitive to intracortical myelin and is not a true measure of tissue myelin content[46,69]. The "cortical layer thickness" maps are from a deep-learning based layer segmentation of the BigBrain histological atlas and are not precise measurements of laminar differentiation of the brain[56–58]. Although we aimed to select non-invasive modalities that are most sensitive to microstructure, cytoarchitecture, and cellular and molecular features, the included maps can only provide proxy, indirect assessments of such biological properties. Finally, despite the fact that we attempt to use a comprehensive list of time-series properties and multiple micro-architectural features, neither the time-series features nor the micro-architectural maps are exhaustive sets of measures. Moreover, micro-architectural features are group-average maps that are compiled from different datasets. Multimodal datasets from the same individuals are required to perform individual-level comparisons between the dynamical and micro-architectural atlases.

Altogether, using a data-driven approach, the present findings show that neurophysiological signatures of cortical micro-architecture are hierarchically organized across the cortex, reflecting the underlying circuit properties. These findings highlight the importance of conventional measures for studying the characteristics of neurophysiological activity, while also identifying less-commonly used time-series features that covary with cortical micro-architecture. Collectively, this work opens new avenues for studying the anatomical basis of neurophysiological activity.

## Methods

### Dataset: human connectome project (HCP)
Resting state magnetoencephalography (MEG) data from a sample of healthy young adults ($n = 33$; age range 22–35 years; 16 female and 17 male) with no familial relationships were obtained from Human Connectome Project (HCP; S900 release[65]; informed consent obtained). The WU-Minn HCP Consortium (consortium of US and European institutions led by Washington University and the University of Minnesota) approved the study protocol. The obtained data includes resting state scans of approximately 6 minutes long (sampling rate = 2034.5 Hz; anti-aliasing low-pass filter at 400 Hz) and empty-room recordings for all participants. 3T structural magnetic resonance imaging (MRI) data and MEG anatomical data (i.e., cortical sheet with 8004 vertices and transformation matrix required for co-registration of MEG sensors and MRI scans) of all participants were also obtained for MEG pre-processing.

### Resting state magnetoencephalography (MEG)
Resting state MEG data was analyzed using `Brainstorm` software, which is documented and freely available for download online under the GNU general public license (see ref. 66; http://neuroimage.usc.edu/brainstorm). For each individual, MEG sensor recordings were registered to their structural MRI scan using the anatomical transformation matrix provided by HCP for co-registration, following the procedure described in `Brainstorm` online tutorials for the HCP dataset (https://neuroimage.usc.edu/brainstorm/Tutorials/HCP-MEG). The data were downsampled to 1/4 of the original sampling rate (i.e., 509 Hz) to facilitate processing. The pre-processing was performed by applying notch filters at 60, 120, 180, 240, and 300 Hz, and was followed by a high-pass filter at 0.3 Hz to remove slow-wave and DC-offset artifacts. Bad channels were marked based on the information obtained through the data management platform of HCP (ConnectomeDB; https://db.humanconnectome.org/). The artifacts (including heartbeats, eye blinks, saccades, muscle movements, and noisy segments) were then removed from the recordings using automatic procedures as proposed by `Brainstorm`. More specifically, electrocardiogram (ECG) and electrooculogram (EOG) recordings were used to detect heartbeats and blinks, respectively. We then used Signal-Space Projections (SSP) to automatically remove the detected

artifacts. We also used SSP to remove saccades and muscle activity as low-frequency (1–7 Hz) and high-frequency (40–240 Hz) components, respectively.

The pre-processed sensor-level data was then used to obtain a source estimation on HCP's fsLR4k cortical surface for each participant (i.e., 8004 vertices). Head models were computed using overlapping spheres and the data and noise covariance matrices were estimated from the resting state MEG and noise recordings. Linearly constrained minimum variance (LCMV) beamforming from `Brainstorm` was then used to obtain the source activity for each participant. We performed data covariance regularization to avoid the instability of data covariance matrix inversion due to the smallest eigenvalues of its eigenspectrum. Data covariance regularization was performed using the "median eigenvalue" method from Brainstorm[66], such that the eigenvalues of the eigenspectrum of data covariance matrix that were smaller than the median eigenvalue were replaced with the median eigenvalue itself. The estimated source variance was also normalized by the noise covariance matrix to reduce the effect of variable source depth. Source orientations were constrained to be normal to the cortical surface at each of the 8004 vertex locations on the fsLR4k surface. Source-level time-series were parcellated into 100 regions using the Schaefer-100 atlas[67] for each participant, such that a given parcel's time series was estimated as the first principal component of its constituting sources' time series. Finally, we estimated source-level signal-to-noise ratio (SNR) as follows[91,92]:

$$\text{SNR} = 10\log_{10}\left(\frac{a^2}{N}\sum_{k=1}^{N}\frac{b_k^2}{s_k^2}\right), \qquad (1)$$

where $a$ is the source amplitude (i.e., typical strength of a dipole, which is 10 nAm[5]), $N$ is the number of sensors, $b_k$ is the signal at sensor $k$ estimated by the forward model for a source with unit amplitude, and $s_k^2$ is the noise variance at sensor $k$. Group-average source-level SNR was parcellated using the Schaefer-100 atlas.

To estimate a measure of environmental and instrumental noise, empty-room MEG recordings of all individuals were obtained from HCP and were pre-processed using an identical procedure to the resting-state recordings. The pre-processed source-level time-series obtained from empty-room recordings were parcellated and subjected to time-series feature extraction analysis to estimate time-series features from noise data for each participant (see *Time-series feature extraction using* `hctsa`).

### Power spectral analysis
Welch's method was used to estimate power spectrum density (PSD) from the source-level time-series for each individual, using overlapping windows of length 4 seconds with 50% overlap. Average power at each frequency band was then calculated for each vertex (i.e., source) as the mean power across the frequency range of a given frequency band. Source-level power data were parcellated into 100 regions using the Schaefer-100 atlas[67] at six canonical electrophysiological bands (i.e., delta ($\delta$: 2–4 Hz), theta ($\theta$: 5–7 Hz), alpha ($\alpha$: 8–12 Hz), beta ($\beta$: 15–29 Hz), low gamma (lo-$\gamma$: 30–59 Hz), and high gamma (hi-$\gamma$: 60–90 Hz)). We contributed the group-average vertex-level power maps on the fsLR4k surface to the publicly available `neuromaps` toolbox[43].

### Intrinsic timescale
The regional intrinsic timescale was estimated using spectral parameterization with the `FOOOF` (fitting oscillations & one over f) toolbox[21]. Specifically, the source-level power spectral density were used to extract the neural timescale at each vertex and for each individual using the procedure described in ref. 30. The `FOOOF` algorithm decomposes the power spectra into periodic (oscillatory) and aperiodic (1/$f$-like) components by fitting the power spectral density in the

log-log space[21] (Supplementary Fig. 2). The algorithm identifies the oscillatory peaks (periodic component), the "knee parameter" $k$ that controls for the bend in the aperiodic component and the aperiodic "exponent" $\chi$[21,30]. The knee parameter $k$ is then used to calculate the "knee frequency" as $f_k = k^{1/\chi}$, which is the frequency where a knee or a bend occurs in the power spectrum density[30]. Finally, the intrinsic timescale $\tau$ is estimated as[30]:

$$\tau = \frac{1}{2\pi f_k}. \qquad (2)$$

We used the FOOOF algorithm to fit the power spectral density with "knee" aperiodic mode over the frequency range of 1–60 Hz. Note that since the first notch filter was applied at 60 Hz during the pre-processing analysis, we did not fit the model above 60 Hz. Following the guidelines from the FOOOF algorithm and Donoghue et al.[21], the rest of the parameters were defined as: peak width limits (peak_width_limits) = 1–6 Hz; maximum number of peaks (max_n_peaks) = 6; minimum peak height (min_peak_height) = 0.1; and peak threshold (peak_threshold) = 2. Intrinsic timescale $\tau$ was estimated at each vertex for each individual and was parcellated using the Schaefer-100 atlas[67]. The performance of model fits by the FOOOF algorithm was quantified as the "goodness of fit" or $R^2$ for each model fitted to a given power spectrum (Supplementary Fig. 2; range of $R^2 = [0.95, 0.98]$). We contributed the group-average vertex-level intrinsic timescale map on the fsLR4k surface to the publicly available neuromaps toolbox[43].

In addition to the aperiodic component used to calculate the intrinsic timescale, the FOOOF spectral parameterization algorithm also provides the extracted peak parameters of the periodic component at each vertex for each participant. We used the oscillatory peak parameters to estimate band-limited power maps that were adjusted for the aperiodic component as opposed to the total power maps estimated above[21] (see *Power spectral analysis*). We defined the power band limits as delta (2–4 Hz), theta (5–7 Hz), alpha (8–14 Hz), and beta (15–30 Hz), based on the distribution of peak center frequencies across all vertices and participants (Supplementary Fig. 2b). Given the lack of clear oscillatory peaks in high frequencies (above 40 Hz), the FOOOF algorithm struggles with detecting consistent peaks in gamma frequencies and above[21,22]. Thus, we did not analyze band-limited power in gamma frequencies using spectral parameterization. For each of the 4 predefined power bands, we estimated an "oscillation score" following the procedure described by Donoghue et al.[21]. Specifically, for each participant and frequency band, we identified the extracted peak at each vertex. If more than one peak was detected at a given vertex, the peak with maximum power was selected. The average peak power was then calculated at each vertex and frequency band across participants. The group-average peak power map was then normalized for each frequency band, such that the average power at each vertex was divided by the maximum average power across all vertices. Separately, we calculated a vertex-level probability map for each frequency band as the percentage of participants with at least one detected peak at a given vertex at that frequency band. Finally, the band-limited "oscillation score" maps were obtained by multiplying the normalized group-average power maps with their corresponding probability maps for each frequency band. The oscillation score maps were parcellated using the Schaefer-100 atlas[67] (Supplementary Fig. 2a).

**Time-series feature extraction using** hctsa
We used the highly comparative time-series analysis toolbox, hctsa[35,36], to perform a massive feature extraction of the pre-processed time-series for each brain region for each participant. The hctsa package extracted over 7000 local time-series features using a wide range of time-series analysis methods[35,36]. The extracted features include, but are not limited to, measures of data distribution, temporal dependency and correlation properties, entropy and variability, parameters of time-series model fit, and nonlinear properties of a given time-series[35,37].

The hctsa feature extraction analysis was performed on the parcellated MEG time-series for each participant. Given that applying hctsa on the full time-series is computationally expensive, we used 80 seconds of data for feature extraction after dropping the first 30 seconds. Previous reports suggest that relatively short segments of about 30 to 120 seconds of resting-state data are sufficient to estimate robust properties of intrinsic brain activity[22]. Nevertheless, to ensure that we can robustly estimate time-series features from 80 seconds of data, we calculated a subset of hctsa features using the catch-22 toolbox[124] on subsequent segments of time-series with varying length for each participant. Specifically, we extracted time-series features from short segments of data ranging from 5 to 125 seconds in increments of 5 s. To identify the time-series length required to estimate robust and stable features, we calculated the Pearson correlation coefficient between features of two subsequent segments (e.g., features estimated from 10 and 5 seconds of data). The correlation coefficient between the estimated features started to stabilize at time-series segments of around 30 s, consistent with previous reports[22] (Supplementary Fig. 12). Following the feature extraction procedure from time-series segments of 80 s, the outputs of the operations that produced errors were removed and the remaining features (6880 features) were normalized across nodes using an outlier-robust sigmoidal transform for each participant separately. A group-average region × feature matrix was generated from the normalized individual-level features. We also applied hctsa analysis to the parcellated empty-room recordings (80 seconds) to estimate time-series features from noise data using an identical procedure to resting-state data, identifying 6148 features per region per participant. The time-series features were normalized across brain regions for each participant. A group-average empty-room feature set was obtained and used for further analysis.

**Micro-architectural features from** neuromaps
We used the neuromaps toolbox (https://github.com/netneurolab/neuromaps)[43] to obtain micro-architectural and neurotransmitter receptor and transporter maps in the maps' native spaces. Details about all maps and their data sources are available in[43]. Briefly, all data that were originally available in any surface space were transformed to the fsLR32k surface space using linear interpolation to resample data and were parcellated into 100 cortical regions using the Schaefer-100 atlas[67] in fsLR32k space. All volumetric data were retained in their native MNI152 volumetric space and were parcellated into 100 cortical regions using the volumetric Schaefer atlas in MNI152 space[67]. Micro-architectural maps included T1w/T2w as a proxy measure of cortical myelin[46,125], cortical thickness[125], principal component of gene expression[44,68], principal component of neurotransmitter receptors and transporters[47], synapse density (using [11C]UCB-J PET tracer that binds to the synaptic vesicle glycoprotein 2A (SV2A))[55,126–137], metabolism (i.e., cerebral blood flow and volume, oxygen and glucose metabolism, glycolytic index)[61], evolutionary and developmental expansion[63], allometric scaling from Philadelphia Neurodevelopmental Cohort (PNC) and National Institutes of Health (NIH)[64]. Neurotransmitter maps included 18 different neurotransmitter receptors and transporters across 9 different neurotransmitter systems, namely serotonin (5-HT1a, 5-HT1b, 5-HT2a, 5-HT4, 5-HT6, 5-HTT), histamine (H3), dopamine (D1, D2, DAT), norepinephrine (NET), acetylcholine ($\alpha 4\beta 2$, M1, VAChT), cannabinoid (CB1), opioid (MOR), glutamate (mGluR5), and GABA (GABAa/bz)[47].

**BigBrain histological data**
Layer thickness data for the 6 cortical layers (I-VI) were obtained from the BigBrain atlas, which is a volumetric, high-resolution

$(20 \times 20 \times 20\ \mu m)$ histological atlas of a post-mortem human brain (65-year-old male)[56–58]. In the BigBrain atlas, sections of the post mortem brain are stained for cell bodies using Merker staining technique[138]. These sections are then imaged and used to reconstruct a volumetric histological atlas of the human brain that reflects neuronal density and soma size and captures the regional differentiation of cytoarchitecture[56–58,107,139]. The approximate cortical layer thickness data obtained from the `BigBrainWarp` toolbox[57], were originally generated using a convolutional neural network that automatically segments the cortical layers from the pial to white surfaces[58]. Full description of how the cortical layer thickness was approximated is available elsewhere[58]. The cortical layer thickness data for the 6 cortical layers were obtained on the *fsaverage* surface (164k vertices) from the `BigBrainWarp` toolbox[57] and were parcellated into 100 cortical regions using the Schaefer-100 atlas[67].

### Cell type-specific gene expression

Regional microarray expression data were obtained from 6 post-mortem brains (1 female, ages 24–57, $42.5 \pm 13.4$) provided by the Allen Human Brain Atlas (AHBA, https://human.brain-map.org; see ref. 44). Data were processed with the abagen toolbox (version 0.1.3-doc; https://github.com/rmarkello/abagen; see ref. 68) using the Schaefer-100 volumetric atlas in MNI space[67].

First, microarray probes were reannotated using data provided by[140]; probes not matched to a valid Entrez ID were discarded. Next, probes were filtered based on their expression intensity relative to background noise[141], such that probes with intensity less than the background in ≥50.00% of samples across donors were discarded. When multiple probes indexed the expression of the same gene, we selected and used the probe with the most consistent pattern of regional variation across donors (i.e., differential stability; see ref. 142), calculated with:

$$\Delta_S(p) = \frac{1}{\binom{N}{2}} \sum_{i=1}^{N-1} \sum_{j=i+1}^{N} \rho[B_i(p), B_j(p)], \qquad (3)$$

where $\rho$ is Spearman's rank correlation of the expression of a single probe, $p$, across regions in two donors $B_i$ and $B_j$, and N is the total number of donors. Here, regions correspond to the structural designations provided in the ontology from the AHBA.

The MNI coordinates of tissue samples were updated to those generated via non-linear registration using the Advanced Normalization Tools (ANTs; https://github.com/chrisfilo/alleninf). To increase spatial coverage, tissue samples were mirrored bilaterally across the left and right hemispheres[143]. Samples were assigned to brain regions in the provided atlas if their MNI coordinates were within 2 mm of a given parcel. If a brain region was not assigned a tissue sample based on the above procedure, every voxel in the region was mapped to the nearest tissue sample from the donor in order to generate a dense, interpolated expression map. The average of these expression values was taken across all voxels in the region, weighted by the distance between each voxel and the sample mapped to it, in order to obtain an estimate of the parcellated expression values for the missing region. All tissue samples not assigned to a brain region in the provided atlas were discarded.

Inter-subject variation was addressed by normalizing tissue sample expression values across genes using a robust sigmoid function[35]:

$$x_{\text{norm}} = \frac{1}{1 + \exp\left(-\frac{(x - \langle x \rangle)}{\text{IQR}_x}\right)}, \qquad (4)$$

where $\langle x \rangle$ is the median and $\text{IQR}_x$ is the normalized interquartile range of the expression of a single tissue sample across genes.

Normalized expression values were then rescaled to the unit interval:

$$x_{\text{scaled}} = \frac{x_{\text{norm}} - \min(x_{\text{norm}})}{\max(x_{\text{norm}}) - \min(x_{\text{norm}})}. \qquad (5)$$

Gene expression values were then normalized across tissue samples using an identical procedure. Samples assigned to the same brain region were averaged separately for each donor and then across donors, yielding a regional expression matrix of 15,633 genes.

Finally, cell type-specific gene expression maps were calculated using gene sets identified by a cell type deconvolution analysis[59,60,70]. Detailed description of the analysis is available at[59]. Briefly, cell-specific gene sets were compiled across 5 single-cell and single-nucleus RNA sequencing studies of adult human post-mortem cortical samples[144–149]. Gene expression maps of the compiled study-specific cell types were obtained from AHBA. Unsupervised hierarchical clustering analysis was used to identify 7 canonical cell classes that included astrocytes (astro), endothelial cells (endo), microglia (micro), excitatory neurons (neuron-ex), inhibitory neurons (neuron-in), oligodendrocytes (oligo) and oligodendrocyte precursors (opc)[59]. We used the resulting gene sets to obtain average cell type-specific expression maps for each of these 7 cell classes from the regional expression matrix of 15,633 genes.

### Partial Least Squares (PLS)

Partial least squares (PLS) analysis was used to investigate the relationship between resting-state MEG time-series features and micro-architecture maps. PLS is a multivariate statistical technique that identifies mutually orthogonal, weighted linear combinations of the original variables in the two datasets that maximally covary with each other, namely the latent variables[80,81]. In the present analysis, one dataset is the `hctsa` feature matrix (i.e., $\mathbf{X}_{n \times t}$) with $n = 100$ rows as brain regions and $t = 6880$ columns as time-series features. The other dataset is the compiled set of micro-architectural maps (i.e., $\mathbf{Y}_{n \times m}$) with $n = 100$ rows (brain regions) and $m = 45$ columns (micro-architecture maps). To identify the latent variables, both data matrices were normalized column-wise (i.e., z-scored) and a singular value decomposition was applied to the correlation matrix $\mathbf{R} = \mathbf{X}'\mathbf{Y}$ as follows:

$$\mathbf{R} = \mathbf{X}'\mathbf{Y} = \mathbf{U}\mathbf{S}\mathbf{V}', \qquad (6)$$

where $\mathbf{U}_{t \times m}$ and $\mathbf{V}_{m \times m}$ are orthonormal matrices of left and right singular vectors and $\mathbf{S}_{m \times m}$ is the diagonal matrix of singular values. Each column of $\mathbf{U}$ and $\mathbf{V}$ matrices corresponds to a latent variable. Each element of the diagonal of $\mathbf{S}$ is the corresponding singular value. The singular values are proportional to the covariance explained by latent variable and can be used to calculate effect sizes as $\eta_i = s_i^2 / \sum_{j=1}^{J} s_j^2$ where $\eta_i$ is the effect size for the $i$-th latent variable (LV$_i$), $s_i$ is the corresponding singular value, and $J$ is the total number of singular values (here $J = m$). The left and right singular vectors $\mathbf{U}$ and $\mathbf{V}$ demonstrate the extent to which the time-series features and micro-architectural maps contribute to latent variables, respectively. Time-series features with positive weights covary with micro-architectural maps with positive weights, while negatively weighted time-series features and micro-architectural maps covary together. Singular vectors can be used to estimate brain scores that demonstrate the extent to which each brain region expresses the weighted patterns identified by latent variables. Brain scores for time-series features and micro-architectural maps are calculated by projecting the original data onto the PLS-derived weights (i.e., $\mathbf{U}$ and $\mathbf{V}$):

$$\text{Brain scores for time} - \text{series features} = \mathbf{X}\mathbf{U}$$
$$\text{Brain scores for micro} - \text{architecture} = \mathbf{Y}\mathbf{V}$$

Loadings for time-series features and micro-architectural maps are then computed as the Pearson correlation coefficient between the original data matrices and their corresponding brain scores. For example, time-series feature loadings are the correlation coefficients between the original `hctsa` time-series feature vectors and PLS-derived brain scores for time-series features.

The statistical significance of latent variables was assessed using 10,000 permutation tests, where the original data was randomized using spatial autocorrelation-preserving nulls (see "Null model" for more details). The PLS analysis was repeated for each permutation, resulting in a null distribution of singular values. The significance of the original singular values were then assessed against the permuted null distributions (Fig. 4a). The reliability of PLS loadings was estimated using bootstrap resampling[150], where rows of the original data matrices **X** and **Y** are randomly resampled with replacement 10,000 times. The PLS analysis was then repeated for each resampled data, generating a sampling distribution for each time-series feature and micro-architectural map (i.e., generating 10,000 bootstrap-resampled loadings). The bootstrap-resampled loading distributions are then used to estimate 95% confidence intervals for loadings (e.g., see Fig. 4d).

Given that PLS-derived brain scores are by design highly correlated, we used a distance-dependent cross-validation analysis to assess the out-of-sample correlations between brain scores[70]. Specifically, 75% of the closest brain regions in Euclidean distance to a random "seed" region were selected as training set, while the 25% remaining distant regions were selected as test set. We then re-ran the PLS analysis on the training set (i.e., 75% of regions) and used the resulting weights (i.e., singular values) to estimated brain scores for test set. The out-of-sample correlation was then calculated as the Spearman's rank correlation coefficient between test set brain scores of time-series features and micro-architectural maps. We repeated this analysis 99 times, such that each time a random brain region was selected as the seed region, yielding distributions of training set brain scores correlations and test set (out-of-sample) correlations (Fig. 4b). Note that 99 is the maximum number of train-test splits here given that brain maps consist of 100 regions.

Finally, we used sparse canonical correlation analysis (sCCA; see ref. 151) as an alternative multivariate analysis technique to assess whether using a different method with sparsity affects the results[151,152]. Similar to PLS, CCA is another reduced-rank regression analysis that is used to identify multivariate linear relationships between two sets of data matrices[70,81,153–155]. The main difference between CCA and PLS is that in CCA the correlation matrix between the input sets is corrected for within-set correlations, ensuring that the identified link between the two input data matrices is not driven by the correlation structure within one of them[81]. Moreover, sparse CCA (sCCA) adds a regularization parameter to the analysis to impose sparsity and avoid overfitting[151]. The regularization parameter ranges between 0 and 1, where 0 corresponds to highest possible sparsity and 1 corresponds to lowest possibility sparsity. We used sCCA (regularization parameter = 0.7) to identify multivariate associations between neurophysiological time-series features and micro-architectural features and found similar results to the original PLS analysis (Supplementary Fig. 5).

### Null model

To make inferences about the topographic correlations between any two brain maps, we implement a null model that systematically disrupts the relationship between two topographic maps but preserves their spatial autocorrelation[73,74,156]. We used the Schaefer-100 atlas in the HCP's fsLR32k grayordinate space[65,67]. The spherical projection of the fsLR32k surface was used to define spatial coordinates for each parcel by selecting the vertex closest to the center-of-mass of each parcel[157–159]. The resulting spatial coordinates were used to generate null models by applying randomly-sampled rotations and reassigning

node values based on the closest resulting parcel (10,000 repetitions). The rotation was applied to one hemisphere and then mirrored to the other hemisphere. Where appropriate, the results were corrected for multiple comparisons by controlling the false discovery rate (FDR correction[160]).

### Reporting summary

Further information on research design is available in the Nature Portfolio Reporting Summary linked to this article.

### Data availability

All data used in the reported analyses are openly available at https://github.com/netneurolab/shafiei_megdynamics. Source data to generate the figures are provided with this paper. The original data used in this study were obtained from the Human Connectome Project (HCP; S900 release) and are publicly available at https://db.humanconnectome.org/. The original HCP data can be accessed following the HCP data use terms. The micro-architectural data is openly available in neuromaps at https://netneurolab.github.io/neuromaps/. The cortical layer thickness data is openly available in BigBrainWarp at https://bigbrainwarp.readthedocs.io/en/latest/. The Allen Human Brain Atlas (AHBA) data is openly available at https://human.brain-map.org. The Schaefer parcellations (i.e., Shcaefer-100 and Schaefer-400 atlases) are openly available at https://github.com/ThomasYeoLab/CBIG/tree/master/stable_projects/brain_parcellation/Schaefer2018_LocalGlobal. Source data are provided with this paper.

### Code availability

Code used to process and analyze data is available on GitHub (https://github.com/netneurolab/shafiei_megdynamics) and on Zenodo (https://doi.org/10.5281/zenodo.8258832[161]). All analyses were conducted using Python 3.7.9, MATLAB R2020a, netneurotools v0.2.3, and other standard Python packages (e.g., Matplotlib, Mayavi, NiBabel, NumPy, Pandas, Scikit-learn, SciPy, Seaborn). MEG data were processed using the open software toolbox Brainstorm v220420 (MATLAB). The open source python toolbox neuromaps v0.0.3 was used to compile the micro-architectural feature maps (Python). Allen Human Brain Atlas (AHBA) data was processed using the abagen toolbox v0.1.3 (Python). BigBrainWarp toolbox was used to obtain the cortical layer thickness data (Python). Time-series analysis was performed using the highly comparative time-series analysis (hctsa) toolbox v1.07 (Matlab). Spectral parameterization of MEG power was performed using the FOOOF toolbox v1.0.0 (Python). PLS analysis was performed using the pyls package (https://github.com/rmarkello/pyls) (Python).

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

## Acknowledgements

We thank Justine Hansen, Estefany Suarez, Filip Milisav, Andrea Luppi, Vincent Bazinet, Zhen-Qi Liu for their comments on the manuscript. B.M. acknowledges support from the Natural Sciences and Engineering Research Council of Canada (NSERC), Canadian Institutes of Health Research (CIHR), Brain Canada Foundation Future Leaders Fund, the Canada Research Chairs Program, the Michael J. Fox Foundation, and the Healthy Brains for Healthy Lives initiative. S.B. acknowledges support from the NIH (R01 EB026299), a Discovery grant from the Natural Science and Engineering Research Council of Canada (NSERC 436355-13), the CIHR Canada research Chair in Neural Dynamics of Brain Systems, the Brain Canada Foundation with support from Health Canada, and the Innovative Ideas program from the Canada First Research Excellence Fund, awarded to McGill University for the Healthy Brains for Healthy Lives initiative. B.V. acknowledges support from NIH National Institute of General Medical Sciences grant (R01GM134363). G.S. acknowledges support from the Natural Sciences and Engineering Research Council of Canada (NSERC) and the Fonds de recherche du Québec - Nature et Technologies (FRQNT).

## Author contributions

Conceptualization: G.S. and B.M.; Methodology: G.S., B.D.F, and B.M.; Formal Analysis: G.S.; Data Curation: G.S.; Writing - Original Draft: G.S. and B.M.; Writing - Review & Editing: G.S., B.D.F, B.V., T.D.S., S.B., and B.M; Visualization: G.S.; Supervision: B.M.

## Competing interests

The authors declare no competing interests.
