## [Peer Review File · Nature Communications]

Neurophysiological signatures of cortical micro-architectureReviewer #1 (Remarks to the Author):

This paper investigates the relationship between spontaneous neurophysiological dynamics across the human brain and cortical micro-architecture. The authors mapped over 6,800 time-series features to a multi-modal atlas and found that neurophysiological dynamics are co-localized with multiple micro-architectural features.

Major comments:

1) The authors use hctsa to generate a massive set of time series features, which may result in features that are difficult to interpret in terms of underlying neurophysiology. For example, one of the top loading time series features from the PLS analysis is the autocorrelation at a lag of 48ms, but it is unclear how to interpret this in terms of neurophysiological activity. I'm guessing that hctsa also generated features capturing the autocorrelation at many lags, so what is the significance of 48ms?

2) I had initially thought that with such a high number of features, the neurophysiology data PCA would be run using time series from each vertex, or at least each subject. I was surprised to find that it was run on a 100 x 6880 matrix. 100 observations seems very small with such a large number of features, and this may be why a single component captured the vast majority of the variance. The fact that the other components explained 10% of the variance or less makes me think that the number of observations was not sufficient to characterize neurophysiological diversity.

3) Given 100 observations, 6880 time series features, and 45 microarchitecture features, I would be extremely surprised if it were impossible to find linear combinations of features from each group that would yield high correlations between them. There doesn't seem to be any guiding theory or model to this approach, and while the authors describe it as data-driven, its scope is so large that there is a danger of veering from data mining to data dredging. The out-of-sample analysis does mitigate this to some extent, but it may be better to use something like sparse PLS (e.g. Chun & Keles, 2010).

<https://www.ncbi.nlm.nih.gov/pmc/articles/PMC2810828/>

4) "We find that PC1 has high spatial correlations with most of those maps ($|r| > 0.36$), and significant correlations with intrinsic timescale ($r_s = 0.84$, $p_{spin} = 0.03$; FDR-corrected) and high gamma ($r_s = 0.87$, $p_{spin} = 0.005$; FDR-corrected)"

The high spatial correlations that are not significant are meaningless. So the traditionally time series features described by PC1 are the intrinsic timescale and high gamma. Is PC1 just reflecting the slope and/or offset of the aperiodic spectrum?

Minor comments:

1) "Given the hierarchical organization of PC1 and its close relationship with power spectral features"

What is the hierarchical organization of PC1? This wasn't clear to me

2) The use of a notch filter restricts the range of frequencies that can be used for the FOOOF fit. What about using iterative zapline to remove line noise? This would allow the full frequency range to be used.

3) The FOOOF algorithm estimates of aperiodic spectral parameters can be sensitive to the choice of the frequency range used. I have found in some datasets that the lower frequency bound has to be about 0.1Hz to get an accurate estimate of the aperiodic slope. I could not find any measures of the model fit reported in the paper.

4) Band-limited power in gamma frequencies was not analyzed using spectral parameterization, but one of the main results from the PCA is that PC1 was highly correlated with high gamma. So what is high gamma? Related to major comment 5 and minor comment 4, does this mean it's just the residual of the slope of a bad $1/f$ fit?

5) What do the time series look like when projected onto PC1? It would be nice to see the power spectral densities after this projection

6) Why were principal components of the gene expression and neurotransmitter receptors and transporters data used in the PLS analysis rather than the raw data? Was it to reduce the number of features? If so, why use 6880 time series features?

Reviewer #2 (Remarks to the Author):

The paper by Shafiei and colleagues details a novel body of work in which the authors have established correlative links between neurophysiological phenomena (predominantly neural oscillations) measured using MEG, and aspects of brain microstructure gathered from many different open source datasets. The headline finding of the paper is that the spatial variation in neurophysiological dynamics is correlated with a number of micro-architectural features, including myeloarchitecture, neurotransmitter receptor density and metabolism.

I believe that this paper is both novel and an important contribution to the literature in this area. Neural oscillations are a well established phenomenon that are known to be involved in critical functionality, including for example, long range connectivity. Moreover neural oscillations are known to be abnormal in a number of psychiatric and neurological problems. However their fundamental origins and relationship to tissue microstructure are poorly understood – this paper sheds significant light on that important question. I am happy to recommend publication. However before publishing, the authors may wish to think about the following comments.

1) The way in which the MEG data are processed seems to me to be quite non-standard; specifically I haven't before come across 'highly comparative time series analysis'. This isn't a criticism! However, because of the way the article is structured with results coming before the detailed methods, the "6880 time series features" appear a little abstract and will likely confuse some readers – particularly those unfamiliar with MEG. I wonder if a paragraph could be added explaining more about what this feature selection process does. Also, from Figure 1, it seems that whilst there may be 6880 features those features cluster making the columns of the "regions x features" matrix highly correlated... would it be possible to better explain what each of the separate clusters in this matrix relate to – I'm aware from the text that some relate to canonical frequencies etc – but a better explanation of this would, in my opinion, help add clarity and impact.

2) For me the "headline" result was in Figure 4D, which showed how the neurophysiological dynamics loads on each of the aspects of microstructure. However I found this quite hard to interpret (admittedly not helped because the bars are colour coded, and I'm colour blind!). I wonder if there is a better way to present this – for example by making a version that simplifies the finding to a bar chart with the 6 summary measures as well as the current plot?

3) In the discussion, the limitations of the MEG aspects of the study were well explained. However I think it would be good to point out limitations in the microstructure metrics; for example, myelin wasn't measured directly but via the ratio of relaxation constants T1 and T2 – whilst a useful indicator this is not a true measure of tissue myelin content. I suspect similar limitations exist on other measurements. I would be tempted to add a little discussion of these limitations, just to ensure a reader is aware of what is being measured directly, and what is inferred based on e.g. imaging etc.

Minor comments:

Strong relationships between tissue myelin and MEG measured signals have been published previously (e.g. Helbling et al, NeuroImage 2015, and Hunt et al, PNAS, 2016 – there may be others). Perhaps referencing these past papers would be helpful?

In the discussion, the authors say that higher SNR measures like iEEG and ECoG may be helpful – however such measures lack whole brain coverage and so it's hard to see how they could be deployed? Wouldn't on scalp MEG be a better fit?

Reviewer #3 (Remarks to the Author):

This manuscript identifies how spatial variations of neurophysiological signals derived from MEG co-localize with a wide set of micro-architecture markers. This study naturally follows previous works from the same research team that explored how molecular markers co-localise with the organization of the human cortex (Hansen et al. 2022, Nature Neuroscience) and with cross-

disorder features (Hansen et al. 2022; Nature Communications), among others.

This study is rigorous and methodologically sound. It expands the previous work with additional microarchitecture maps and, more importantly, explores in a meticulous way, thousands of features derived from MEG dynamics. As in previous studies, I have to congratulate the team for the effort in providing data and code that is curated and ready to use. The sensitivity section already addressed the only methodological concerns that I initially had so I recommend this paper for publication.

Thank you for the constructive feedback on our initial submission and for the opportunity to revise and resubmit the manuscript. Following the Reviewers' comments and suggestions, we have thoroughly revised the manuscript. In this letter, we respond to each of the Reviewers' comments in detail. Reviewer comments are in bold font and our responses are in regular font.

Reviewer #1 (Remarks to the Author):

This paper investigates the relationship between spontaneous neurophysiological dynamics across the human brain and cortical micro-architecture. The authors mapped over 6,800 time-series features to a multi-modal atlas and found that neurophysiological dynamics are co-localized with multiple micro-architectural features.

Major comments:

1) The authors use *hctsa* to generate a massive set of time series features, which may result in features that are difficult to interpret in terms of underlying neurophysiology. For example, one of the top loading time series features from the PLS analysis is the autocorrelation at a lag of 48ms, but it is unclear how to interpret this in terms of neurophysiological activity. I'm guessing that *hctsa* also generated features capturing the autocorrelation at many lags, so what is the significance of 48ms?

We thank the Reviewer for their comment. Yes, that is correct. The *hctsa* toolbox generates features capturing the autocorrelation at a range of time lags. Specifically, the studied range comprises 40 time lags between 1 and 40 time steps, corresponding to 2 ms and 80 ms, respectively. We selected a time lag of 48 ms to provide an example of the topographic distribution of top-loading features in the analysis (absolute loading of AC-48ms = 0.71). However, the top loading features also include other time lags with similar or higher loadings. The figure below shows the PLS loadings for all time lags that were included in the analysis:

To clarify this, we have now included this figure as a supplementary figure in the revised manuscript. We also include all the information related to *hctsa* feature loadings in machine-readable format in Supplementary File S2. In addition, we have added the following text to the revised manuscript (“Results” section, “Neurophysiological signatures of micro-architecture” subsection, paragraph #4):

“Another example is the linear autocorrelation of the MEG signal at longer time lags. Fig.4c, *left* shows autocorrelation at a lag of 48 ms (24 time steps), demonstrating lower autocorrelation in unimodal cortex and higher autocorrelation in transmodal cortex. Note that the list of top-loading features includes linear autocorrelation at other time lags and autocorrelation at a lag of 48 ms was only selected as an illustrative example (Fig. S5 depicts the full range of loadings for linear autocorrelation at all time lags included in *hctsa*).”

And (“Results” section, “Neurophysiological signatures of micro-architecture” subsection, paragraph #3):

“The full list of time-series feature loadings for the first latent variable is available in the online Supplementary File S2.”

2) I had initially thought that with such a high number of features, the neurophysiology data PCA would be run using time series from each vertex, or at least each subject. I was surprised to find that it was run on a 100 x 6880 matrix. 100 observations seems very small with such a large number of features, and this may be why a single component captured the vast majority of the variance. The fact that the other components explained 10% of the variance or less makes me think that the number of observations was not sufficient to characterize neurophysiological diversity.

We concur with the Reviewer that projecting a small number of observations to a redundantly high dimensional space is associated with various issues in statistical analysis (also known as “curse of dimensionality”). However, to our knowledge, the smaller number of samples than features should not per se drive the amount of variance explained by the principal components extracted by PCA, given that PCA by design reduces the data dimensionality while preserving the maximum variation in the data regardless of the number of features or samples (Jolliffe & Cadima, 2016). Nevertheless, to ensure that the findings were independent from the data dimensionality, we performed a sensitivity analysis where we randomly selected 100 *hctsa* features from the original 6880 features and re-ran PCA on the new 100 x 100 matrix (rows correspond to regions and columns correspond to randomly selected *hctsa* features). We then computed the correlation coefficient between the PC scores derived from the PCA applied to the smaller 100 x 100 matrix and those from the original PCA applied to the larger 100 x 6880 matrix. We repeated this procedure 1000 times.

The figure below depicts the variance explained by PC1, PC2, and PC3 as well as the Pearson correlation coefficients between the PCs from randomly selected 100 x 100 matrices and the

ones from the original analysis. We note that the variance explained from the original analysis (shown by vertical red lines) is consistent with the distribution of variance explained obtained from the sensitivity analysis (shown by blue histograms) for all PCs. Moreover, the spatial distributions of PC scores are highly correlated between the original and sensitivity analysis (shown by box plots below the histograms; PC1: $r > 0.99$, PC2: $r > 0.8$, PC3: $r > 0.75$).

We have modified the text and added the figure below to the revised manuscript to reflect this analysis (“Results” section, “Topographic distribution of neurophysiological dynamics” subsection, paragraph #4):

“Moreover, to verify that the apparent low-dimensionality of the data and the identified PC patterns were not driven by the smaller number of samples (i.e., brain regions) than features (i.e., time-series features), we performed a sensitivity analysis where we randomly selected 100 time-series features (from the original list of 6880 features) and re-ran PCA (1000 repetitions). The identified PC patterns and their corresponding amount of variance explained were consistent with the original analysis using the full set of time-series features (Fig. S3).”

In addition, we repeated the analysis with a cortical atlas with a higher parcellation resolution to assess whether the findings were affected by the number of parcels (i.e. number of observations or samples). Specifically, we used the Schaefer-400 atlas with 400 cortical regions instead of the Schaefer-100 atlas with 100 regions used in the original analysis. The results were consistent for both parcellations. We have included this analysis in the manuscript along with a supplementary figure depicting the results (“Results” section, “Sensitivity analysis” subsection, paragraph #2):

“Finally, to ensure that the findings are independent from the parcellation resolution, we repeated the analyses using a higher resolution parcellation (Schaefer-400 atlas with 400 cortical regions (Schaefer et al., 2018)). The results were consistent with the original analysis (Fig. S9 and Fig. S10). In particular, the first principal component (PC1) accounted for 48.6% of the variance and displayed a similar spatial organization as the one originally obtained for the Schaefer-100 atlas (Fig. S9a). As before, the top loading time-series features were mainly related to the characteristics of the power spectral density (Fig. S9b,c). The full list of features, their loadings and p -values are available in the online Supplementary File S3.”

Jolliffe, I. T., & Cadima, J. (2016). Principal component analysis: a review and recent developments. *Philosophical transactions of the royal society A: Mathematical, Physical and Engineering Sciences*, 374(2065), 20150202.

3) Given 100 observations, 6880 time series features, and 45 microarchitecture features, I would be extremely surprised if it were impossible to find linear combinations of features from each group that would yield high correlations between them. There doesn't seem to be any guiding theory or model to this approach, and while the authors describe it as data-driven, its scope is so large that there is a danger of veering from data mining to data dredging. The out-of-sample analysis does mitigate this to some extent, but it may be better to use something like sparse PLS (e.g. Chun & Keles, 2010).

<https://www.ncbi.nlm.nih.gov/pmc/articles/PMC2810828/>

We concur with the Reviewer that multivariate associative analyses (such as PLS) will, by design, maximize the correlation between linear combinations of features. We also agree that the data dimensionality makes it challenging to reliably identify such associations. Our main goal in this study was to go beyond a small number of manually selected time-series and micro-architectural features-of-interest. Instead, we attempted to comprehensively map extensive sets of dynamical and micro-architectural features across multiple scales using data-driven approaches. Such all-to-all mapping approaches using multivariate analysis methods have been previously used in the literature, providing useful charts on how multimodal brain data are associated to each other (Hansen et al., 2021) or how they relate to behavioral and/or clinical phenotypes (Smith et al., 2015; Avants et al., 2014; Xia et al., 2018; Kirschner et al., 2020).

However, to mitigate challenges that accompany data-driven analysis with multivariate approaches, our original PLS analysis involved a rigorous out-of-sample cross-validation testing (as pointed out by the Reviewer). We also performed a sensitivity analysis using a higher resolution parcellation with 400 cortical regions (Figure S10). Moreover, following the Reviewer's suggestion, we repeated the analysis using a Python implementation of sparse canonical correlation analysis (sCCA; Witten et al., 2009) to investigate the multivariate associations between time-series and microarchitectural features. We opted to use sCCA given that it is commonly used in the field, making the analysis technique comparable to previous

reports (Smith et al., 2015; Avants et al., 2014; Xia et al., 2018; Hansen et al., 2021). Another advantage of using CCA is that in CCA the correlation matrix between the input sets is corrected for within-set correlations (as opposed to PLS), ensuring that the identified link between the two input data matrices is not driven by the correlation structure within one of them (McIntosh & Mišić, 2013). We used sCCA to relate MEG time-series features to microarchitectural features and found similar findings to the original PLS analysis. The figure below shows the findings from sCCA:

We have included this new analysis in the revised manuscript and have added a new supplementary figure to show the results:

“Results” section, “Neurophysiological signatures of micro-architecture”, paragraph #1:

“Fig.4a shows that the analysis identifies multiple such combinations, termed latent variables (similar results were obtained using sparse canonical correlation analysis (sCCA); Fig. S4).”

“Methods” section, “Partial Least Squares (PLS)” subsection, paragraph #5:

“Finally, we used sparse canonical correlation analysis (sCCA; Witten et al., 2009) as an alternative multivariate analysis technique to assess whether using a different method with sparsity affects the results (Witten et al., 2009; Chun & Keleş, 2010). Similar to PLS, CCA is another reduced-rank regression analysis that is used to identify multivariate linear relationships between two sets of data matrices (McIntosh & Mišić, 2013; Smith et al., 2015; Avants et al., 2014; Xia et al., 2018; Hansen et al., 2021). The main difference between CCA and PLS is that in CCA the correlation matrix between the input sets is corrected for within-set correlations, ensuring that the identified link between the two input data matrices is not driven by the correlation structure within one of them (McIntosh & Mišić, 2013). Moreover, sparse CCA (sCCA) adds a regularization parameter to the analysis to impose sparsity and avoid overfitting (Witten et al., 2009). The regularization parameter ranges between 0 and 1, where 0 corresponds to highest possible sparsity and 1 corresponds to lowest possibility sparsity. We used sCCA (regularization parameter = 0.7) to identify multivariate associations between neurophysiological time-series features and micro-architectural features and found similar results to the original PLS analysis (Fig. S4).”

In addition, we added the following sentence to the Introduction section in the revised manuscript (“Introduction” section, paragraph #4):

“Instead of manually selecting a small number of features-of-interest, we use extensive sets of dynamical and micro-architectural features using data-driven approaches.”

Witten, D. M., Tibshirani, R., & Hastie, T. (2009). A penalized matrix decomposition, with applications to sparse principal components and canonical correlation analysis. *Biostatistics*, 10(3), 515-534.

Chun, H., & Keleş, S. (2010). Sparse partial least squares regression for simultaneous dimension reduction and variable selection. *Journal of the Royal Statistical Society: Series B (Statistical Methodology)*, 72(1), 3-25.

McIntosh, A. R., & Mišić, B. (2013). Multivariate statistical analyses for neuroimaging data. *Annual review of psychology*, 64, 499-525.

Smith, S. M., Nichols, T. E., Vidaurre, D., Winkler, A. M., Behrens, T. E., Glasser, M. F., ... & Miller, K. L. (2015). A positive-negative mode of population covariation links brain connectivity, demographics and behavior. *Nature neuroscience*, 18(11), 1565-1567.

Avants, B. B., Libon, D. J., Rascovsky, K., Boller, A., McMillan, C. T., Massimo, L., ... & Grossman, M. (2014). Sparse canonical correlation analysis relates network-level atrophy to multivariate cognitive measures in a neurodegenerative population. *Neuroimage*, 84, 698-711.

Xia, C. H., Ma, Z., Ciric, R., Gu, S., Betzel, R. F., Kaczkurkin, A. N., ... & Satterthwaite, T. D. (2018). Linked dimensions of psychopathology and connectivity in functional brain networks. *Nature communications*, 9(1), 3003.

Kirschner, M., Shafiei, G., Markello, R. D., Makowski, C., Talpalaru, A., Hodzic-Santor, B., ... & Mišić, B. (2020). Latent clinical-anatomical dimensions of schizophrenia. *Schizophrenia bulletin*, 46(6), 1426-1438.

Hansen, J. Y., Markello, R. D., Vogel, J. W., Seidlitz, J., Bzdok, D., & Misic, B. (2021). Mapping gene transcription and neurocognition across human neocortex. *Nature Human Behaviour*, 5(9), 1240-1250.

4) "We find that PC1 has high spatial correlations with most of those maps ($|r| > 0.36$), and significant correlations with intrinsic timescale ($r_s = 0.84$, $p_{\text{spin}} = 0.03$; FDR-corrected) and hi-gamma ($r_s = 0.87$, $p_{\text{spin}} = 0.005$; FDR-corrected)"

The high spatial correlations that are not significant are meaningless. So the traditionally time series features described by PC1 are the intrinsic timescale and high gamma. Is PC1 just reflecting the slope and/or offset of the aperiodic spectrum?

We concur with the Reviewer that reporting the non-significant correlations is not useful. We have now removed this information and modified the text in the revised manuscript ("Results" section, "Topographic distribution of neurophysiological dynamics" subsection, paragraph #3):

"We find that PC1 is significantly correlated with intrinsic timescale ($r_s = 0.84$, $p_{\text{spin}} = 0.03$; FDR-corrected) and hi-gamma ($r_s = 0.87$, $p_{\text{spin}} = 0.005$; FDR-corrected)."

We thank the Reviewer for their comment about the slope and offset of the aperiodic component of the power spectrum. Our analysis demonstrates that PC1 captures the linear correlation structure of the neurophysiological activity and broad variations in its power spectrum. These properties are also reflected in intrinsic timescale, which is directly estimated from the knee frequency (f_k) and exponent (χ) of the aperiodic component of the power spectrum. More specifically, intrinsic timescale is estimated as: $\tau = 1/2\pi f_k$, where:

- $f_k = k^{1/\chi}$ is the knee frequency, which is the frequency at which a knee or a bend occurs in the power spectrum density,
- k is the knee parameter, which controls for the bend in the aperiodic component,
- χ is the aperiodic exponent.

Hence, as the Reviewer points out, PC1 reflects characteristics of the aperiodic component of the power spectrum, which are also mathematically related to the intrinsic timescale. To directly assess the association between PC1 and the aperiodic component of the power spectrum, we

correlated PC1 with the exponent (slope) and the offset of the aperiodic component. As expected, PC1 was significantly correlated with both measures (Figure below).

We have now modified the manuscript to reflect this analysis (“Results” section, “Topographic distribution of neurophysiological dynamics” subsection, paragraph #3):

“Given that the intrinsic timescale reflects characteristics of the aperiodic component of the power spectrum (these measures are mathematically related; see *Methods* for details), we also directly assessed the association between PC1 and the exponent and offset of the aperiodic component. The exponent describes the “curve” or the overall “line” or the slope of the aperiodic component and the offset describes the overall vertical shift (up and down translation) of the whole power spectrum (Donoghue et al., 2020). PC1 was significantly correlated with both measures, suggesting that time-series features captured by PC1 also reflect properties of the aperiodic component of the power spectrum (Fig. S2).”

Minor comments:

1) "Given the hierarchical organization of PC1 and its close relationship with power spectral features"

What is the hierarchical organization of PC1? This wasn't clear to me

We concur with the Reviewer that the phrasing of this sentence is unclear. The original sentence referred to the topographic organization of PC1 that varied along an axis from dorsal attention, somatomotor and visual networks to limbic and default mode networks. However, to avoid any confusion, we have now rephrased this sentence in the revised manuscript (“Results” section, “Topographic distribution of neurophysiological dynamics” subsection, paragraph #3):

“Given that the topographic organization of PC1 was closely related to power spectral features, we directly tested the link between PC1 and conventional band-limited power spectral measures (Donoghue et al., 2020; Wiesman et al., 2022; da Silva et al., 2021), as well as the intrinsic timescale (Gao et al., 2020).”

2) The use of a notch filter restricts the range of frequencies that can be used for the FOOOF fit. What about using iterative zapline to remove line noise? This would allow the full frequency range to be used.

We would like to note that FOOOF can still fit a wide range of frequencies even when a notch filter is applied. This can be achieved by FOOOF by "ignoring" a frequency band to skip over line noise frequencies or by interpolating line noise regions. However, FOOOF struggles with reliably detecting oscillatory peaks in high frequencies (above 40 Hz), given the lack of clear power peaks in those ranges (Donoghue et al., 2020; Wiesman et al., 2022). Hence, to avoid any overfitting and to only focus on clear and reliably detectable oscillatory peaks, we used the FOOOF algorithm and spectral parameterization for frequencies up to 40 Hz (as recommended by FOOOF as well). The main goal of this analysis was to ensure that our original findings with total power (Figure 3) were consistent with the results obtained from the power maps adjusted for the aperiodic activity (Figure S1) (at least in the frequency range with clear oscillatory peaks).

We have included an explanation on the choice of frequency range for the FOOOF algorithm in the manuscript ("Methods" section, "Intrinsic timescale" subsection, paragraph #3):

"Given the lack of clear oscillatory peaks in high frequencies (above 40 Hz), the FOOOF algorithm struggles with detecting consistent peaks in gamma frequencies and above (Donoghue et al., 2020; Wiesman et al., 2022). Thus, we did not analyze band-limited power in gamma frequencies using spectral parameterization."

Donoghue, T., Haller, M., Peterson, E. J., Varma, P., Sebastian, P., Gao, R., ... & Voytek, B. (2020). Parameterizing neural power spectra into periodic and aperiodic components. *Nature neuroscience*, 23(12), 1655-1665.

Wiesman, A. I., da Silva Castanheira, J., & Baillet, S. (2022). Stability of spectral estimates in resting-state magnetoencephalography: Recommendations for minimal data duration with neuroanatomical specificity. *Neuroimage*, 247, 118823.

3) The FOOOF algorithm estimates of aperiodic spectral parameters can be sensitive to the choice of the frequency range used. I have found in some datasets that the lower frequency bound has to be about 0.1Hz to get an accurate estimate of the aperiodic slope. I could not find any measures of the model fit reported in the paper.

We thank the Reviewer for pointing this out. In our analysis, the R^2 of the model fit (goodness of the fit) ranged between 0.95 and 0.98 (Figure below), suggesting that the FOOOF algorithm provided good model fits to the power spectra.

We have now updated the supplementary figure 1 (Figure S1) to include the distribution of R^2 values. We have also updated the Methods section to reflect this information (“Methods” section, “Intrinsic timescale” subsection, paragraph #2):

“The performance of model fits by the FOOOF algorithm was quantified as the “goodness of fit” or R^2 for each model fitted to a given power spectrum (Fig. S1; range of $R^2 = [0.95, 0.98]$.”

a | spectral parameterization of neurophysiological activity

b | distribution of center frequencies

c | PC1 score of *hctsa*

4) Band-limited power in gamma frequencies was not analyzed using spectral parameterization, but one of the main results from the PCA is that PC1 was highly correlated with high gamma. So what is high gamma? Related to major comment 5 and minor comment 4, does this mean it's just the residual of the slope of a bad $1/f$ fit?

The high gamma power included in the main analysis (e.g., as shown in Figure 3) is the average power in the range of 60-90 Hz, obtained from the total power spectrum without adjusting for the aperiodic activity. We have now clarified this in the revised manuscript (“Methods” section, “Power spectral analysis” subsection, paragraph #1):

“Average power at each frequency band was then calculated for each vertex (i.e., source) as the mean power across the frequency range of a given frequency band.”

However, as mentioned above in response to the Reviewer's minor comment 2, we used the FOOOF algorithm as a sensitivity analysis to ensure that our original findings with total power

(Figure 3) were consistent with the results obtained from the power spectrum adjusted for the aperiodic activity (Figure S1). The sensitivity analysis with the spectral parameterization using the FOOOF algorithm was only performed in the frequency range with clear oscillatory peaks. Specifically, since the oscillatory peaks in high frequencies (above 40 Hz) are not clear enough to be reliably detected by the FOOOF algorithm, we used spectral parameterization for frequencies up to 40 Hz as recommended by the FOOOF algorithm (Donoghue et al., 2020; Wiesman et al., 2022). In other words, we used the FOOOF algorithm to estimate band-limited power maps that were adjusted for the aperiodic component of the power spectrum, focusing on the frequency range with clear and reliably detectable peaks.

We have now included an explanation on the choice of the frequency range for the FOOOF algorithm in the manuscript ("Methods" section, "Intrinsic timescale" subsection, paragraph #3):

"Given the lack of clear oscillatory peaks in high frequencies (above 40 Hz), the FOOOF algorithm struggles with detecting consistent peaks in gamma frequencies and above (Donoghue et al., 2020; Wiesman et al., 2022). Thus, we did not analyze band-limited power in gamma frequencies using spectral parameterization."

Furthermore, following the Reviewer's comment, we have modified the manuscript to include further explanation on the high correlation between PC1 and high gamma power from the original analysis. As mentioned above in our response to major comment 4 of the Reviewer, PC1 is associated with intrinsic timescale and directly reflects properties of the aperiodic component of the power spectrum. In line with this finding, previous reports have suggested that broadband gamma activity partly reflects the aperiodic neurophysiological activity and broadband shifts in the power spectrum (Manning et al., 2009; Buzsáki & Wang, 2012; Hudson & Jones, 2022). This is consistent with our findings that PC1 is associated with gamma power and intrinsic timescale, mainly capturing broad variations in power spectrum and characteristics of the aperiodic activity. We have modified the manuscript to reflect this explanation ("Results" section, "Topographic distribution of neurophysiological dynamics" subsection, paragraph #3):

"In addition, previous reports suggest that broadband gamma activity also partly reflects the aperiodic neurophysiological activity and broadband shifts in the power spectrum (Manning et al., 2009; Buzsáki & Wang, 2012; Hudson & Jones, 2022). This is consistent with our findings that PC1 is associated with gamma power and intrinsic timescale, mainly capturing broad variations in power spectrum and characteristics of the aperiodic activity."

Manning, J. R., Jacobs, J., Fried, I., & Kahana, M. J. (2009). Broadband shifts in local field potential power spectra are correlated with single-neuron spiking in humans. *Journal of Neuroscience*, 29(43), 13613-13620.

Buzsáki, G., & Wang, X. J. (2012). Mechanisms of gamma oscillations. *Annual review of neuroscience*, 35, 203-225.

Donoghue, T., Haller, M., Peterson, E. J., Varma, P., Sebastian, P., Gao, R., ... & Voytek, B. (2020). Parameterizing neural power spectra into periodic and aperiodic components. *Nature neuroscience*, 23(12), 1655-1665.

Wiesman, A. I., da Silva Castanheira, J., & Baillet, S. (2022). Stability of spectral estimates in resting-state magnetoencephalography: Recommendations for minimal data duration with neuroanatomical specificity. *Neuroimage*, 247, 118823.

Hudson, M. R., & Jones, N. C. (2022). Deciphering the code: Identifying true gamma neural oscillations. *Experimental Neurology*, 114205.

5) What do the time series look like when projected onto PC1? It would be nice to see the power spectral densities after this projection

If we understand the Reviewer's comment correctly, the question is how the PC patterns would be organized spatially if the time-series were projected onto PC1. We apologize in advance if the Reviewer refers to something else and that we are missing the Reviewer's point. However if we understand correctly, we note that our analysis is slightly different, such that we first replace time-series with time-series features extracted using *hctsa*. We then apply PCA to the feature-based representation of the time-series (rather than to the time-series themselves). As a result, PC weights are estimated for time-series features rather than time points. This will not allow us to project the time-series onto PC1 and other PCs. Instead, we project the time-series feature onto PC1 and obtain PC1 spatial distribution as depicted in the manuscript (Figure 2).

6) Why were principal components of the gene expression and neurotransmitter receptors and transporters data used in the PLS analysis rather than the raw data? Was it to reduce the number of features? If so, why use 6880 time series features?

We included principal components of gene expression and neurotransmitter receptors and transporters as they each represent proxy measures of certain molecular properties. Specifically, the principal component of gene expression (gene expression PC1) provides a potential proxy for cell type distribution across the cortex (Hawrylycz et al., 2012; Burt et al., 2018; Hansen et al., 2021) and the principal component of neurotransmitter receptors and transporters (neurotransmitter PC1) provides a summary measure of protein densities of 18 neurotransmitter receptors and transporters (Hansen et al., 2022a; Hansen et al., 2022b). However, we also included individual neurotransmitter receptor and transporter maps as well as cell type-specific gene expression maps to assess their effects separately.

We have now clarified this in the revised manuscript ("Results" section, paragraph #2):

“Note that the microstructure maps include principal gradients of gene expression and neurotransmitter profiles as they each represent proxy measures of certain molecular properties. Specifically, the principal component of gene expression (gene expression PC1) provides a potential proxy for cell type distribution across the cortex (Hawrylycz et al., 2012; Burt et al., 2018; Hansen et al., 2021) and the principal component of neurotransmitter receptors and transporters (neurotransmitter PC1) provides a summary measure of protein densities of 18 neurotransmitter receptors and transporters (Hansen et al., 2022a; Hansen et al., 2022b). We also included individual neurotransmitter receptor and transporter maps as well as cell type-specific gene expression maps to assess their effects separately.”

Hawrylycz, M. J., Lein, E. S., Guillozet-Bongaarts, A. L., Shen, E. H., Ng, L., Miller, J. A., ... & Jones, A. R. (2012). An anatomically comprehensive atlas of the adult human brain transcriptome. *Nature*, 489(7416), 391-399.

Burt, J. B., Demirtaş, M., Eckner, W. J., Navejar, N. M., Ji, J. L., Martin, W. J., ... & Murray, J. D. (2018). Hierarchy of transcriptomic specialization across human cortex captured by structural neuroimaging topography. *Nature neuroscience*, 21(9), 1251-1259.

Hansen, J. Y., Markello, R. D., Vogel, J. W., Seidlitz, J., Bzdok, D., & Misić, B. (2021). Mapping gene transcription and neurocognition across human neocortex. *Nature Human Behaviour*, 5(9), 1240-1250.

Hansen, J. Y., Shafiei, G., Markello, R. D., Smart, K., Cox, S. M., Nørgaard, M., ... & Misić, B. (2022). Mapping neurotransmitter systems to the structural and functional organization of the human neocortex. *Nature Neuroscience*, 1-13.

Hansen, J. Y., Shafiei, G., Voigt, K., Liang, E., Cox, S. M., Leyton, M., ... & Misić, B. (2022). Multimodal, multiscale connectivity blueprints of the cerebral cortex. *bioRxiv*, 2022-12.

Reviewer #2 (Remarks to the Author):

The paper by Shafiei and colleagues details a novel body of work in which the authors have established correlative links between neurophysiological phenomena (predominantly neural oscillations) measured using MEG, and aspects of brain microstructure gathered from many different open source datasets. The headline finding of the paper is that the spatial variation in neurophysiological dynamics is correlated with a number of micro-architectural features, including myeloarchitecture, neurotransmitter receptor density and metabolism.

I believe that this paper is both novel and an important contribution to the literature in this area. Neural oscillations are a well established phenomenon that are known to be involved in critical functionality, including for example, long range connectivity. Moreover neural oscillations are known to be abnormal in a number of psychiatric and neurological problems. However their fundamental origins and relationship to tissue microstructure are poorly understood – this paper sheds significant light on that important question. I am happy to recommend publication. However before publishing, the authors may wish to think about the following comments.

1) The way in which the MEG data are processed seems to me to be quite non-standard; specifically I haven't before come across 'highly comparative time series analysis'. This isn't a criticism! However, because of the way the article is structured with results coming before the detailed methods, the "6880 time series features" appear a little abstract and will likely confuse some readers – particularly those unfamiliar with MEG. I wonder if a paragraph could be added explaining more about what this feature selection process does. Also, from Figure 1, it seems that whilst there may be 6880 features those features cluster making the columns of the "regions x features" matrix highly correlated... would it be possible to better explain what each of the separate clusters in this matrix relate to – I'm aware from the text that some relate to canonical frequencies etc – but a better explanation of this would, in my opinion, help add clarity and impact.

We have now expanded our explanation on *hctsa* time-series features of MEG data in the beginning of the Results section. Specifically, we have included the passage below in the revised manuscript ("Results" section, paragraph #1):

"Highly comparative time-series analysis (*hctsa*; (Fulcher et al., 2013; Fulcher & Jones, 2017)) was then used to perform massive time-series feature extraction from regional MEG recordings. This procedure provides a feature-based representation of time-series, where given time-series are represented by time-series feature vectors (Fulcher & Jones, 2017; Fulcher, 2018). This time-series phenotyping analysis is a data-driven method that quantifies dynamic repertoire of neural activity using interdisciplinary metrics of temporal structure of the signal and yields a comprehensive 'fingerprint' of dynamical properties of each brain region. Applying time-series phenotyping to regional MEG time-series, we

estimated 6880 time-series features for 100 cortical regions from the Schaefer-100 atlas (Schaefer et al., 2018). The *hctsa* library contains a vast and interdisciplinary set of time-series features with potentially correlated values that span various conceptual properties. The list of time-series features includes, but is not limited to, statistics derived from the autocorrelation function, power spectrum, amplitude distribution, and entropy estimates (Fig. 1).”

We have also added the following explanation regarding correlated time-series features (“Results” section, “Topographic distribution of neurophysiological dynamics”, paragraph #1):

“Since *hctsa* contains multiple algorithmic variants for quantifying any given time-series property, the identified time-series features potentially capture related dynamical behaviour and include groups of correlated properties. Hence, we first sought to identify dominant macroscopic patterns or gradients of neurophysiological dynamics using principal component analysis (PCA) (Shafiei et al., 2020).”

2) For me the “headline” result was in Figure 4D, which showed how the neurophysiological dynamics loads on each of the aspects of microstructure. However I found this quite hard to interpret (admittedly not helped because the bars are colour coded, and I’m colour blind!). I wonder if there is a better way to present this – for example by making a version that simplifies the finding to a bar chart with the 6 summary measures as well as the current plot?

We thank the Reviewer for bringing this to our attention. We have now made 3 adjustments to Figure 4 to address this issue:

- (1) We have changed the categorical colormap used to color code the microarchitectural subsets and the corresponding bar charts. We now use the colorblind-friendly colormap from the Python package Seaborn (https://seaborn.pydata.org/tutorial/color_palettes.html), which displays reasonably better with colorblind filters (although unfortunately it is still not perfect).
- (2) Following the Reviewer’s suggestion, we have also modified the bar charts in Figure 4D, only displaying the maps with reliable loadings (i.e., with confidence intervals that do not cross zero). This helps simplify the bar chart. We have also tilted the x-axis labels to make them more readable.
- (3) Following the Reviewer’s other suggestion, we have now included a subplot to Figure 4D, depicting the findings in the bar chart with 6 summary measures (Figure 4D, *right*). Specifically, we plot the loadings for each of the 6 microarchitectural categories as scattered points, showing the distribution of loadings for microarchitectural maps.

We have also modified Figure 1 and Figure S10 (and the new Figure S4) to reflect the new colormap. We thank the Reviewer for flagging this important issue!

a | effect size

b | brain scores

c | example top loading time-series features

d | microarchitecture loadings

3) In the discussion, the limitations of the MEG aspects of the study were well explained. However I think it would be good to point out limitations in the microstructure metrics; for example, myelin wasn't measured directly but via the ratio of relaxation constants T1 and T2 – whilst a useful indicator this is not a true measure of tissue myelin content. I suspect similar limitations exist on other measurements. I would be tempted to add a little discussion of these limitations, just to ensure a reader is aware of what is being measured directly, and what is inferred based on e.g. imaging etc.

We concur with the Reviewer that microarchitectural maps are by no means direct measurements of the underlying microstructure, cytoarchitecture, and cellular and molecular features. At best, they provide proxy measures that are indirectly related to such biological properties. We have now added a new point to the Discussion section to acknowledge this limitation (“Discussion” section, paragraph #6):

“Third, we note that the included micro-architectural maps are by no means direct measurements of the underlying neurobiological features. For example, the “myelin” map is estimated based on the ratio of T1-weighted to T2-weighted MRI scans, which is only sensitive to intracortical myelin and is not a true measure of tissue myelin content (Glasser et al., 2011, Burt et al., 2018). The “cortical layer thickness” maps are from a deep-learning based layer segmentation of the BigBrain histological atlas and are not precise measurements of laminar differentiation of the brain (Amunts et al., 2013; Paquola et al., 2021; Wagstyl et al., 2020). Although we aimed to select non-invasive modalities that are most sensitive to microstructure, cytoarchitecture, and cellular and molecular features, the included maps can only provide proxy, indirect assessments of such biological properties.”

Glasser, M. F., & Van Essen, D. C. (2011). Mapping human cortical areas in vivo based on myelin content as revealed by T1- and T2-weighted MRI. *Journal of neuroscience*, 31(32), 11597-11616.

Burt, J. B., Demirtaş, M., Eckner, W. J., Navejar, N. M., Ji, J. L., Martin, W. J., ... & Murray, J. D. (2018). Hierarchy of transcriptomic specialization across human cortex captured by structural neuroimaging topography. *Nature neuroscience*, 21(9), 1251-1259.

Amunts, K., Lepage, C., Borgeat, L., Mohlberg, H., Dickscheid, T., Rousseau, M. É., ... & Evans, A. C. (2013). BigBrain: an ultrahigh-resolution 3D human brain model. *science*, 340(6139), 1472-1475.

Paquola, C., Royer, J., Lewis, L. B., Lepage, C., Glatard, T., Wagstyl, K., ... & Bernhardt, B. (2021). The BigBrainWarp toolbox for integration of BigBrain 3D histology with multimodal neuroimaging. *Elife*, 10, e70119.

Wagstyl, K., Larocque, S., Cucurull, G., Lepage, C., Cohen, J. P., Bludau, S., ... & Evans, A. C. (2020). BigBrain 3D atlas of cortical layers: Cortical and laminar thickness gradients diverge in sensory and motor cortices. *PLoS biology*, 18(4), e3000678.

Minor comments:

Strong relationships between tissue myelin and MEG measured signals have been published previously (e.g. Helbling et al, NeuroImage 2015, and Hunt et al, PNAS, 2016 – there may be others). Perhaps referencing these past papers would be helpful?

We thank the Reviewer for pointing this out. We have now referenced these two papers in the revised manuscript (“Discussion” section, paragraph #5):

“Our findings build on previous reports by showing that neurophysiological dynamics follow the underlying cytoarchitectonic and microstructural gradients. In particular, our findings confirm that MEG intrinsic dynamics are associated with the heterogeneous distribution of gene expression and intracortical myelin (Gao et al., 2020; Demirtas et al., 2019; Helbling et al., 2015; Hunt et al., 2016) and neurotransmitter receptors and transporters (Hansen et al., 2022).”

Helbling, S., Teki, S., Callaghan, M. F., Sedley, W., Mohammadi, S., Griffiths, T. D., ... & Barnes, G. R. (2015). Structure predicts function: Combining non-invasive electrophysiology with in-vivo histology. *Neuroimage*, 108, 377-385.

Hunt, B. A., Tewarie, P. K., Mougín, O. E., Geades, N., Jones, D. K., Singh, K. D., ... & Brookes, M. J. (2016). Relationships between cortical myeloarchitecture and electrophysiological networks. *Proceedings of the National Academy of Sciences*, 113(47), 13510-13515.

In the discussion, the authors say that higher SNR measures like iEEG and ECoG may be helpful – however such measures lack whole brain coverage and so its hard to see how they could be deployed? Wouldn't on scalp MEG be a better fit?

We thank the Reviewer for their comment. We have modified the manuscript to reflect this point (“Discussion” section, paragraph #6):

“Second, MEG is susceptible to low SNR and has variable sensitivity to neural activity from different regions (i.e., sources). Thus, electrophysiological recordings with higher spatial resolution, such as intracranial electroencephalography (iEEG and ECoG), may provide more precise measures of neural dynamics that can be examined with respect to cortical micro-architecture. However, a major caveat with iEEG and ECoG is that they lack whole brain coverage, limiting their practical usage in such analysis. An alternative non-invasive modality is on-scalp MEG, which offers both high SNR and spatial resolution (Boto et al., 2016; Pfeiffer et al., 2018; Tierney et al., 2019; Hill et al., 2020).”

Boto, E., Bowtell, R., Krüger, P., Fromhold, T. M., Morris, P. G., Meyer, S. S., ... & Brookes, M. J. (2016). On the potential of a new generation of magnetometers for MEG: a beamformer simulation study. *PLoS one*, 11(8), e0157655.

Pfeiffer, C., Andersen, L. M., Lundqvist, D., Hämäläinen, M., Schneiderman, J. F., & Oostenveld, R. (2018). Localizing on-scalp MEG sensors using an array of magnetic dipole coils. *PLoS One*, 13(5), e0191111.

Tierney, T. M., Holmes, N., Mellor, S., López, J. D., Roberts, G., Hill, R. M., ... & Barnes, G. R. (2019). Optically pumped magnetometers: From quantum origins to multi-channel magnetoencephalography. *NeuroImage*, 199, 598-608.

Hill, R. M., Boto, E., Rea, M., Holmes, N., Leggett, J., Coles, L. A., ... & Brookes, M. J. (2020). Multi-channel whole-head OPM-MEG: Helmet design and a comparison with a conventional system. *NeuroImage*, 219, 116995.

Reviewer #3 (Remarks to the Author):

This manuscript identifies how spatial variations of neurophysiological signals derived from MEG co-localize with a wide set of micro-architecture markers. This study naturally follows previous works from the same research team that explored how molecular markers co-localise with the organization of the human cortex (Hansen et al. 2022, *Nature Neuroscience*) and with cross-disorder features (Hansen et al. 2022; *Nature Communications*), among others.

This study is rigorous and methodologically sound. It expands the previous work with additional microarchitecture maps and, more importantly, explores in a meticulous way, thousands of features derived from MEG dynamics. As in previous studies, I have to congratulate the team for the effort in providing data and code that is curated and ready to use. The sensitivity section already addressed the only methodological concerns that I initially had so I recommend this paper for publication.

We thank the Reviewer for their kind words!

Reviewer #1 (Remarks to the Author):

I would like to thank the authors for their thorough and insightful replies to my comments. I am happy to recommend the manuscript for publication.

- James Bonaiuto

Reviewer #2 (Remarks to the Author):

The authors have now addressed all of my concerns and I believe that this paper should be published.

My sincere congratulations to the authors on a really nice piece of work!